# An invasive appetite: Combining molecular and stable isotope analyses to reveal the diet of introduced house mice (*Mus musculus*) on a small, subtropical island

**Wieteke A. Holthuijzen**[1]*, **Elizabeth N. Flint**[2], **Stefan J. Green**[3], **Jonathan H. Plissner**[4], **Daniel Simberloff**[1], **Dagmar Sweeney**[5], **Coral A. Wolf**[6], **Holly P. Jones**[7,8]

1 Department of Ecology & Evolutionary Biology, University of Tennessee at Knoxville, Knoxville, Tennessee, United States of America, 2 Marine National Monuments of the Pacific, U.S. Fish and Wildlife Service, Honolulu, Hawaiʻi, United States of America, 3 Genomics and Microbiome Core Facility, Rush University Medical Center, Chicago, Illinois, United States of America, 4 Midway Atoll National Wildlife Refuge, U.S. Fish and Wildlife Service, Waipahu, Hawaiʻi, United States of America, 5 Institute for Health Research & Policy, University of Illinois at Chicago, Chicago, Illinois, United States of America, 6 Island Conservation, Santa Cruz, California, United States of America, 7 Department of Biological Sciences, Northern Illinois University, DeKalb, Illinois, United States of America, 8 Institute for the Study of the Environment, Sustainability, and Energy, Northern Illinois University, DeKalb, Illinois, United States of America

* wholthuijzen@gmail.com

**Data Availability Statement:** All relevant data are available at Dryad and are accessible at the

## Abstract

House mice (*Mus musculus*) pose a conservation threat on islands, where they adversely affect native species' distributions, densities, and persistence. On Sand Island of Kuaihelani, mice recently began to depredate nesting adult mōlī (Laysan Albatross, *Phoebastria immutabilis*). Efforts are underway to eradicate mice from Sand Island, but knowledge of mouse diet is needed to predict ecosystem response and recovery following mouse removal. We used next-generation sequencing to identify what mice eat on Sand Island, followed by stable isotope analysis to estimate the proportions contributed by taxa to mouse diet. We collected paired fecal and hair samples from 318 mice between April 2018 to May 2019; mice were trapped approximately every eight weeks among four distinct habitat types to provide insight into temporal and spatial variation. Sand Island's mice mainly consume arthropods, with nearly equal (but substantially smaller) contributions of $C_3$ plants, $C_4$ plants, and mōlī. Although seabird tissue is a small portion of mouse diet, mice consume many detrital-feeding arthropods in and around seabird carcasses, such as isopods, flesh flies, ants, and cockroaches. Additionally, most arthropods and plants eaten by mice are non-native. Mouse diet composition differs among habitat types but changes minimally throughout the year, indicating that mice are not necessarily limited by food source availability or accessibility. Eradication of house mice may benefit seabirds on Sand Island (by removing a terrestrial, non-native predator), but it is unclear how arthropod and plant communities may respond and change. Non-native and invasive arthropods and plants previously consumed (and possibly suppressed) by mice may be released post-eradication, which could prevent recovery of native taxa. Comprehensive knowledge of target species' diet is a critical component of eradication planning. Dietary information should be used both to identify and to

following link: https://doi.org/10.5061/dryad.q573n5tnn. The data file in Dryad contains five spreadsheets: House Mouse Metadata; Plant ASV Table; Arthropod ASV Table; House Mouse SIA Data; and Diet Item SIA Data.

**Funding:** This research was funded the National Science Foundation Graduate Research Fellowship Program (DG#-1842161; https://www.nsfgrfp.org/; WAH), Northern Illinois University Elwood and Ruth Briles Memorial Fund (https://niu.academicworks.com/opportunities/17418; WAH), Sigma Xi Grants in Aid of Research (G20200315103593991; https://www.sigmaxi.org/programs/grants-in-aid-of-research; WAH), and Island Conservation (https://www.islandconservation.org/; WAH). Bioinformatics analysis in this article was performed by the University of Illinois at Chicago (UIC) Research Informatics Core, supported in part by the National Center for Advancing Translational Sciences (NCATS) through Grant No. UL1TR002003. The funding sources financially supported the data collection and analysis in this manuscript but did not have any other involvement in the study design, decision to publish, or preparation of the manuscript. Any opinions, findings, and conclusions or recommendations expressed in this material are those of the authors and do not necessarily reflect the views of the National Science Foundation, U.S. Fish and Wildlife Service, or Island Conservation. Mention of trade names or commercial products does not constitute their endorsement.

**Competing interests:** The authors have declared that no competing interests exist.

monitor which taxa may respond most strongly to invasive species removal and to assess if proactive, pre-eradication management activities are warranted.

## Introduction

House mice (*Mus musculus*) are among the most widespread invasive species on islands and have recently been recognized as a major threat to insular biodiversity. Invasive mice attack and prey upon eggs, chicks, and adult birds on islands, leading to population declines in threatened seabirds and landbirds [1–5]. More broadly, mice can modify invertebrate, plant, and soil communities via predation, vegetation damage, and soil loss and even alter ecosystem processes, such as nutrient cycling [6–9]. In turn, predation by invasive mice has prompted eradication efforts where their impacts have been especially severe.

Currently, a house mouse (hereafter "mouse") eradication is planned for Sand Island, where mice were discovered consuming adult, incubating mōlī (Laysan Albatross, *Phoebastria immutabilis*) in 2015 [10]. Sand Island is part of Kuaihelani, a small, subtropical atoll in the north Pacific Ocean that contains the largest mōlī nesting colony in the Pacific Ocean, providing breeding grounds for 73% of this albatross' global population [10, 11]. Although mouse removal will benefit mōlī, it is unknown which other species are affected by mice and how they will respond. More specifically, eradication efforts may release taxa that were consumed and possibly suppressed by mice. While this would be a favorable result for previously-consumed native species, such a release of invasive species could hamper the broader recovery of Sand Island's ecosystem. Comprehensive knowledge of mouse diet is therefore crucial to identify potential adverse consequences of eradication operations and outcomes.

Previous research of house mice on islands shows that mice have complex, omnivorous diets but tend to prefer arthropods (especially Lepidopteran larvae), followed by seeds and vegetative materials [2, 5, 6, 9, 12–24]. Beyond arthropods and plants, mice have been documented to consume a variety of other foods, including seabirds (via depredation and/or scavenging; [1, 4, 10, 16, 25–27]), landbirds [28], skinks, and possibly other reptiles [29]. Only one study has noted the presence of fungi and spores of vesicular-arbuscular mycorrhizal fungi in mouse diet [30], although it is unclear if mice were actively preying on these fungi, or if they were inadvertently consumed while mice were eating plant roots. While these studies have contributed important information about mouse diet and ecology, nearly all document challenges owing to diet determination methods used. For example, some studies mention biases in identifying prey species (i.e., microhistological surveys; [17]) or make assumptions about mouse diet in estimates of food source contributions (i.e., stable isotope analysis; [19, 21]). Combining dietary determination methods—especially for cryptic, understudied taxa, such as insular house mouse populations—is a more informative and robust approach that can reduce biases and increase taxonomic resolution [31]. In particular, advanced molecular techniques and stable isotope analysis are promising complementary methods [32, 33]. Next-generation sequencing (NGS) of gut contents and fecal matter has produced increasingly powerful analyses that can identify food taxa with high taxonomic resolution [34–41]. Subsequently, NGS data can inform stable isotope analysis, which quantifies the diet proportion contributed by food taxa through indirect methods (such as Bayesian mixing models; [42]).

Here, we ascertain the diet of mice on Sand Island using next-generation sequencing (NGS) and stable isotope analysis (SIA). Specifically, we identify taxa eaten by mice via NGS; in turn, we use NGS data to determine which taxa to collect for SIA and include in our stable isotope mixing models to determine taxa contributions to overall mouse diet. In addition, we examine

differences in taxa consumed by mice among habitat types and trapping sessions to assess spatial and temporal dietary variability. We predict that:

1. Arthropods will constitute the majority of mouse diet on Sand Island, given mouse preference for arthropods [23]. Mice will favor large, slow-moving arthropods [9], specifically Blattodea (cockroaches), Coleoptera (beetles), Orthoptera (crickets and grasshoppers), and Lepidoptera (moths and butterflies).

2. Plants will contribute to mouse diet to a lesser degree; mice will prefer seed and vegetative material from grasses and sedges [6, 23].

3. Mōlī will be a minor food source, as mouse aggression towards mōlī has been observed on Sand Island only during winter [10], possibly when other food sources are scarce.

4. Mice will have distinct trends in diet composition among habitat types and sampling sessions, affected by small home ranges and phenological shifts in prey availability and accessibility throughout the year (e.g., breeding seabirds).

## Materials and methods

### Study site

Kuaihelani (also known as Pihemanu, Midway Atoll National Wildlife Refuge [NWR]; 28°11′41″–28°16′50″ N and 177°18′38″–177°25′38″ W) is located near the end of the Northwestern Hawaiian Islands, a 2,400-kilometer chain of protected islands, atolls, coral reefs, and seamounts in the north Pacific Ocean (Fig 1A). Three coral islands compose the atoll system of Kuaihelani (Fig 1B): Sand (457.7 ha), Eastern (136.4 ha), and Spit (5.1 ha; [43]). Kuaihelani has a subtropical climate with relatively even temperatures year-round and pronounced dry/wet seasons [10]. Island interiors are characterized by salt-tolerant and drought-resistant forbs, grasses, and vines, surrounded by a perimeter of coastal scrub vegetation [44]. Most notably, Kuaihelani supports globally-significant avian populations; over three million birds from >25 species breed on or migrate through Kuaihelani each year [10, 11]. However, this atoll system has also undergone extensive landscape modification and disturbance throughout the 20th century. In particular, Sand Island contains large stands of introduced trees (namely *Casuarina* spp.) and other plants, as well as large, barren areas created during wartime eras, including runways, concrete foundations, asphalt, buildings, piers, and other structures [45, 46]. Additionally, house mice and black rats (*Rattus rattus*) were inadvertently introduced to Kuaihelani in 1943 (during World War II) when they escaped from U.S. Navy ships [46, 47]. Rats were successfully eradicated from both Sand and Eastern Islands in 1996 [48], but mice remain on Sand Island.

### Sample collection

For this study, we received samples from mice captured by the U.S. Fish and Wildlife Service (USFWS) and Island Conservation (IC) via eight traplines across Sand Island to account for differences in food source composition, availability, and accessibility (Fig 1B). Each trapline consisted of five pre-baited traps with peanut butter and oats (Trapper 24/7: Bell Laboratories, Inc., Windsor, USA) spaced 10 m apart, for a total of 40 traps (*n* = 40); these traps are capable of multiple captures and were designed specifically for house mice. Mice were trapped across four habitat types: forest, herbland, shrub, and wetland (S1 Table). Seven trapping sessions occurred between April 2018 and May 2019 (Fig 1B); each averaged 6.1 (± 0.7 SD) trapping days (or 5.1 ± 0.7 SD trap nights), and intervals between trapping sessions averaged 55.8 (±

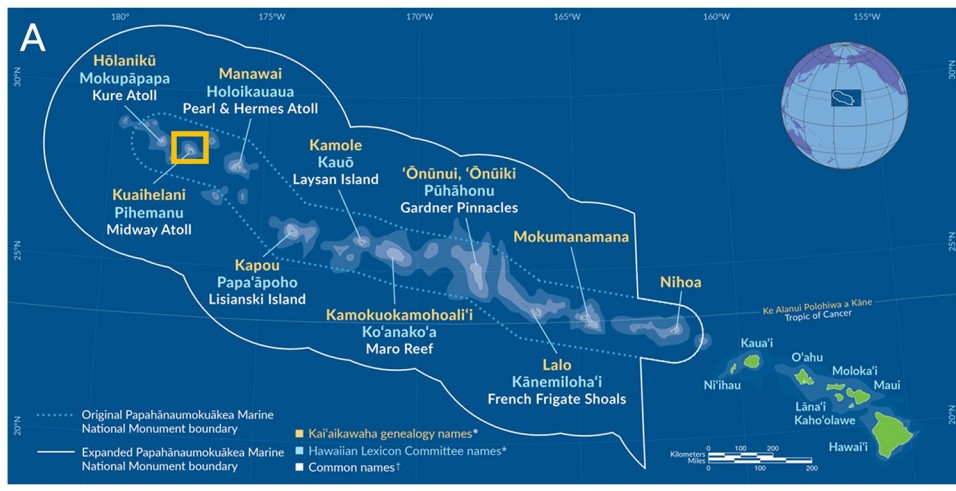

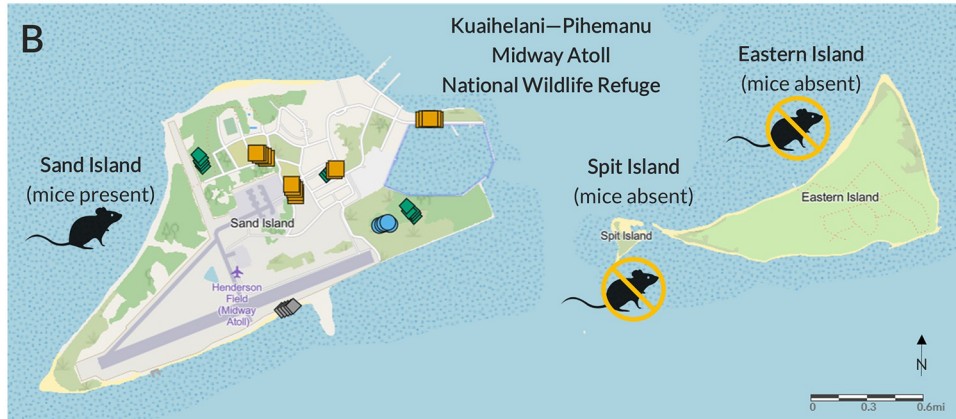

| Habitat Type | Number of Traps | | Trapping Session | Dates |
|---|---|---|---|---|
| ◆ Forest | 13 | | 1 | April 29—May 4, 2018 |
| ◼ Herbland | 17 | | 2 | June 17—22, 2018 |
| ◆ Shrub | 5 | | 3 | July 29—August 3, 2018 |
| ● Wetland | 5 | | 4 | October 1—5, 2018 |
| *Total* | *40* | | 5 | November 28—December 3, 2018 |
| | | | 6 | February 2—8, 2019 |
| | | | 7 | April 29—May 5, 2019 |

**Fig 1. Study site location and overview of house mouse trapping efforts on Sand Island of Kuaihelani.** A) Map of the Hawaiian archipelago in the north Pacific Ocean, including the Papahānaumokuākea Marine National Monument; Kuaihelani is outlined in orange (map credit: NOAA [public domain]). B) Aerial view of Kuaihelani and house mouse trap locations (vector basemap data © OpenStreetMap contributors, hosted by Esri [under CC BY 4.0]).

15.0 SD) days. For each mouse captured, the following data were collected: sex (female, male, unknown), reproductive condition (if female), and age (juvenile, adult, unknown). USFWS and IC staff euthanized mice via cervical dislocation and then removed colon contents and hair (collected from the lower abdomen area) in a sterilized lab setting. Because Sand Island is a remote field station and lacks appropriate laboratory facilities, colon samples were individually stored in vials of 70% isopropyl alcohol, labeled, and placed in a -20°C freezer until shipment to the mainland for further preparation and processing; hairs were kept in dry, individually-sealed vials at room temperature.

In total, USFWS and IC staff trapped 806 mice from April 2018 to May 2019 (S2 Table). Of these, 606 mice composed our sampling pool of adult female and male mice. We did not include female mice in a reproductive state (i.e., lactating, pregnant, contained embryos, etc.), juvenile mice, or mice with an unconfirmed age or sex. We sought a minimum sample size of ≥10 adult mice (pooled female and male) per habitat type, per trapping session to conduct robust statistical analyses (i.e., 10 mice x 4 habitat types x 7 trapping sessions = 280 mice). To maximize our coverage of potential spatial and temporal variation in diet, we randomly selected 318 mice for dietary analyses—the greatest number of mice that we were able to process based on logistical and financial constraints (S2 Table).

House mice are managed as an invasive species on Kuaihelani (Midway Atoll NWR); trapping, euthanizing, and processing mice on Kuaihelani falls under the jurisdiction and management authorizations of USFWS. Neither USFWS nor IC (partner organization with the pending mouse eradication) require an institutional animal care and use committee (IACUC) for invasive house mouse removal and related work. Nevertheless, both organizations follow IACUC guidelines (e.g., checking mouse traps daily and early and euthanizing the animals via cervical dislocation at the trapping site to minimize suffering [49]).

### Next-generation sequencing (NGS)

**Extraction.**   Upon receiving colon samples from Sand Island, we dissected the samples to obtain fecal pellets for DNA extraction in a sterilized lab setting using flame-sanitized tools with 95% ethyl alcohol before each dissection. Fecal pellets were stored in 2 mL individually-labeled vials at -80°C. Genomic DNA was extracted with a DNeasy PowerSoil Pro kit (Qiagen, Carlsbad, USA) implemented on a QIAcube Instrument (Qiagen), following the manufacturer's protocol. Samples were homogenized with a FastPrep-24 System (MP Biomedicals, Irvine, USA) for 40 s at 6 m/s.

**Sequencing and library preparation.**   Genomic DNA was prepared for sequencing with a two-stage amplicon sequencing workflow [50]. Initially, genomic DNA was PCR-amplified using two primer sets (independently). For arthropods, we used the fwhF2/fwhR2n primers to target a 205 bp region of cytochrome c oxidase I (COI) of the *MT-CO1* gene [51, 52]. For plants, we used the UniplantF/UniplantR primers to target a 187–387 bp region of the internal transcribed spacer 2 (ITS2) region of nuclear ribosomal DNA (between genes *5.8S* and *28S*; [53]). The primers—fwhF2/fwhR2n and UniplantF/UniplantR—contained 5' linker sequences compatible with Access Array primers for Illumina sequencers (Fluidigm, South San Francisco, USA). PCRs were performed in a total volume of 10 μL using MyTaq™ HS 2X Mix (Bioline/Meridian Bioscience, Cincinnati, USA), with primers at 500 nM. We also attempted to amplify genomic DNA with several bird-specific primers but were unsuccessful owing to high host DNA amplification, despite the use of peptide-nucleic acid (PNA) oligonucleotide blockers and restriction enzymes. For more details about PCR conditions and outcomes, see S1 Appendix.

DNA extraction, library preparation, pooling, and sequencing were performed at the University of Illinois at Chicago (UIC) Genome Research Core within the Research Resources Center. For plant amplicons, DNA sequencing was performed on an Illumina MiSeq instrument (San Diego, California) employing V3 chemistry (600 cycles, 2 x 300 bp). For arthropod amplicons, DNA sequencing took place on an Illumina NovaSeq 6000 instrument with an SP flow cell type (2 x 250 bp), owing to high host DNA amplification.

**Bioinformatics pipeline.**   Bioinformatics analysis was performed by the UIC Research Informatics Core within the Research Resources Center. Forward and reverse reads were merged using paired-end read merger (PEAR) [54]. For sequencing trimming, we used

cutadapt, and quality trimming was based on quality threshold and length parameters (p = 20, min length = 100 bp; [55]). Adapter/primer sequences were trimmed from reads; reads that lacked the adapter/primer sequences were also discarded. Additionally, ambiguous nucleotides (N) were trimmed from the ends and reads with internal ambiguous nucleotides were discarded. Chimeric sequences are artifacts of the PCR process and occur when portions of two separate amplicons fuse during the amplification process. The UIC analysis pipeline uses a standard chimera checking program to identify chimeric sequences and remove them from datasets; chimeric sequences were identified using USEARCH in a denovo fashion [56]. We then generated separate amplicon sequence variant (ASV) tables for plant and arthropod amplicons using the DADA2 pipeline [57] with a 97% identity (sequence similarity) threshold in order to increase accuracy of taxonomic assignment and exclude chimeric sequences. Sequences were matched to references (thereby producing taxonomic annotations) using a nucleotide BLAST search [58] via nt, the NCBI Genbank non-redundant nucleotide database [59]. We then exported these data along with raw sequence counts into tables of taxonomic annotations from phylum- to species-level results (separately for plant and arthropod ASVs). For more details about sequence data, see S1 Appendix.

## Stable isotope data

**Tissue collection.** For stable isotope analysis (SIA), we used mouse hair samples because they are simple to obtain and store but also because of the unique growth cycle of mouse hair. Unlike that of other mammals, mouse hair does not grow at a continuous rate. Instead, mice grow a pelt or coat (once—maybe twice—depending on longevity), which is not continuously growing. However, single hair follicles do undergo successive growth phases—namely growth (anagen), regression (catagen), and rest (telogen)—that may take up to seven weeks to complete [60, 61]. In addition, not all hairs are shed during these growth cycles, and up to four hairs (all of which may be in different phases) are retained in one follicle [60]. Thus, mouse hair follicles are in a continuum of growth phases [62], making it difficult to ascertain the time during which a hair was actively growing and—in turn—assimilating carbon and nitrogen isotopes from food. For our project, we used only paired colon and hair samples from mice identified as "adult" (i.e., >7 weeks old), meaning that they had completed at least one full hair cycle (and perhaps more). Accordingly, the use of mouse hairs for SIA allows us to understand broader trends in mouse diet because hair samples likely contain hairs in multiple growth phases representing several weeks to months of food assimilation.

Based on the plant and arthropod ASV tables generated from NGS, we collected samples of plant and arthropod taxa detected in ≥5% of the mouse fecal samples (*sensu* [32]). Plant samples were obtained between April 2020 to November 2021. We sought to collect ≥10 samples per taxon, including leaves, stems, inflorescences, fruits, and seeds. Plant samples were air-dried at room temperature for one day, then stored in individually-labelled coin envelopes in a plastic bag filled with silica beads. In addition to mouse trapping for this study, we also completed arthropod pitfall trapping simultaneously along the same traplines (from April 2018 to May 2020; see [63]). Arthropod specimens collected from these pitfall traps were used for SIA. As with plant taxa, we sought to collect ≥10 samples per arthropod taxon. For larger arthropods, such as Araneae and Blattodea, we used only the head and legs for SIA; however, for smaller arthropods, such as Isopoda and Hymenoptera, we used the entire body as well as one or more individuals in a sample to obtain an appropriate sample mass for SIA. Arthropods from pitfall traps were stored in 70% ethanol. We also collected mōlī feathers from January through November in 2019, based on the documented predatory and consumptive behavior of mice on Sand Island [10]. Feather samples were stored in individually-labeled coin envelopes

at room temperature. Because mouse diet on Sand Island was unknown prior to our NGS work, it was not possible to sample food taxa for SIA until *after* mouse colon and hair samples had already been collected and analyzed; regardless, we attempted to collect food taxa from different habitats throughout the year to account for spatial and temporal variation in isotopic values.

**Stable isotope analysis.** To prepare samples for stable isotope analysis, we soaked samples in a 2:1 mixture of methanol/chloroform for at least two hours in an analog orbital shaker at a low-shake frequency (200 rpm) to remove any contaminants or residue [64]. We then rinsed all samples twice in DI water and oven-dried them at 55˚C for 72 hours. After drying, samples were homogenized and weighed to the nearest 0.002 mg. Prior to packing our samples into tin capsules, we used a mortar and pestle (which was cleaned before and after each sample) to crush arthropod and plant samples and we cut mouse hairs and mōlī feathers into the smallest pieces possible. We used approximately 0.8 mg of mouse hairs, 1.0 mg of mōlī feathers, 3.0 mg of plant tissue, and 1.0 mg of arthropod tissue to analyze for carbon and nitrogen elemental composition (C:N ratio) and carbon and nitrogen isotopes ($\delta^{13}$C and $\delta^{15}$N). We processed our samples in isotope-ratio mass spectrometers (Delta Plus Advantage Mass Spectrometer and Delta Plus XL Mass Spectrometer, Thermo Fisher Scientific, Waltham, USA) at Northern Illinois University (NIU) and the University of Tennessee–Knoxville (UTK). Stable isotope ratios are expressed in $\delta$ notation in per mil units (‰), based on the equation:

$$\delta X = \left( \left[ \frac{R_{sample}}{R_{standard}} \right] - 1 \right) \times 1000$$

where $X$ is $^{13}$C or $^{15}$N and $R$ is the corresponding ratio of $^{13}$C to $^{12}$C or $^{15}$N to $^{14}$N. $R_{standard}$ values are based on Vienna PeeDee Belemnite (VPDB) for $\delta^{13}$C and atmospheric $N_2$ for $\delta^{15}$N. At NIU, samples were compared against NBS 22, USGS-25, IAEA-CH-6, IAEA-N1, and IAEA-N2 standards. At UTK, samples were corrected to the standard values of UT729, acet-Cost, and LGlu4510; isotopic values were calibrated against the international standard materials of USGS40 and USGS41. Sample precision based on repeated standard reference materials was ±0.1‰ for $\delta^{13}$C and ±0.2‰ for $\delta^{15}$N; sample precision between the NIU and UTK laboratories based on replicate tissue samples was ±0.5‰ for $\delta^{13}$C and ±0.4‰ for $\delta^{15}$N.

## Statistical analyses

**Sequence data analysis.** For our plant ASV table, we filtered out all ASVs assigned to any phylum other than Streptophyta (land plants and all green algae except Chlorophyta). Similarly, for the arthropod ASV table, we filtered out all ASVs assigned to any phylum other than Arthropoda. For both plants and arthropods, we used only ASVs identified at least to genus in our subsequent analyses. For plant ASVs, we removed *Arachis* sp. and *Avena* sp. because we used peanut butter and oats as bait in house mouse traps, which occasionally were detected in mouse fecal samples (but neither species occurs on Kuaihelani).

Because of biases that occur during PCR, sequencing, and bioinformatic analysis processes, sequence read counts provide—at best—a coarse estimate of how much a given ASV contributes to diet. In other words, the percentage of sequence reads belonging to each food taxon is not a reliable proxy for relative biomass consumed [41, 65]. Thus, we converted ASV sequence read counts to occurrence data (i.e., presence/absence of food taxa). Although this transformation can introduce biases—such as omitting poorly amplified taxa [66]—it is a more conservative, semi-quantitative diet summary. Although we sought to retain as many ASVs as possible (including those with low read counts), we used both sequence read normalization and read thresholds to exclude artifacts. We obtained the relative read abundance (RRA—proportional

summaries of sequence read counts; [66]) for each food taxon by dividing the number of reads per ASV by the total count of ASVs for each sample, separately for both the plant and arthropod ASV tables. We considered a taxon to be present if it contributed ≥1% of the sequence reads per sample [66, 67]; 1% is considered a suitable threshold for many diet studies, but lower thresholds may be required when diets are extremely diverse or large and variable differences occur in amplification of food taxa [68]. We used these transformed occurrence datasets (%FOO—percent frequency of occurrence; [66]) to summarize diet taxa and for additional statistical analyses.

We conducted multivariate analyses using PRIMER-e v7 [69] to examine spatial and temporal differences in plant and arthropod consumption by mice. First, we computed two resemblance matrices of occurrence data (plant and arthropod ASVs) using the Sørensen dissimilarity index [70] and visually examined the output using principal coordinates analysis (PCoA; [69, 71]). We then used permutational multivariate analysis of variance (PERMANOVA) tests with habitat type and trapping session as fixed effects in a two-way crossed design using a Type III SS (with 9,999 iterations). Here, the estimated sizes of components of variation can be used to compare the relative importance of different effects in a model to explain overall variation [71]. Significant results from a PERMANOVA test indicate differences in plant or arthropod consumption among groups (i.e., centroid location effect) and/or variability within groups (i.e., dispersion effect; [72]). As a follow-up, we used the PERMDISP test to check for differences in the homogeneity of multivariate dispersion among habitat type and trapping session groups [71]; in other words, PERMDISP tests the variability of plant and arthropod composition in diet among groups. Significant results from PERMDISP help to elucidate if plant or arthropod consumption differs owing to distinct diet composition and/or variability in diet composition.

**Dietary mixing model analysis.** When selecting sources for our mixing model analysis, we retained only plant and arthropod taxa detected in ≥5% of mouse fecal samples (i.e., % FOO ≥5%), so as to avoid overparameterizing our model with rare food sources (*sensu* [32, 73]). Additionally, even though we were unable to detect mōlī DNA in mouse fecal samples, we decided to include mōlī as a source as well (given mouse predatory behavior on Sand Island; [4, 10]). In total, we identified 25 food taxa from ASV data as sources for our model; however, with so many sources—and only two isotopic tracers—a mixing model is unlikely to yield precise estimates of source proportion in diet [73]. To reduce the number of sources initially, we *a priori* categorized food taxa into three biologically-meaningful and taxonomically-relevant groups: $C_3$ plants, $C_4$ plants, and mōlī. For the remaining arthropod taxa, we used Ward's hierarchical cluster analysis ("ward.D2" from the "hclust" function in R package 'stats'; [74]) and multivariate tests (PERMANOVA and PERMDISP in PRIMER-e; [69]) to determine the most appropriate grouping of arthropods into source groups. Through cluster analysis, we sought to categorize arthropods based on feeding ecology [75] while minimizing isotopic overlap among groups. However, many of the arthropod taxa consumed by Sand Island's mice are generalist omnivores, detritivores, and scavengers and share similar isotopic signatures (S3 Table). Five source groups emerged via cluster analysis, namely: 1) two Lepidopteran taxa (phytophagous larvae and adults); 2) *Megaselia scalaris* (Diptera; zoophagous, detritophagous, and mycetophagous larvae; predator, parasitoid); 3) *Periplaneta* sp. (Blattodea; detritophagous larvae and adults); 4) *Trachyzelotes jaxartensis* (Araneae; predator); and 5) an array of isotopically-similar taxa from Diptera, Ixodida, Hymenoptera, and Isopoda with diverse feeding ecologies (S3 Table). For each of these arthropod source groups, within-group isotopic variation was smaller than among-group variation. Thus, our stable isotope mixing model had eight main source groups: $C_3$ plants, $C_4$ plants, Diptera-Ixodida-Hymenoptera-Isopoda, Lepidoptera, *M. scalaris*, mōlī, *Periplaneta* sp., and *T. jaxartensis* (S1 Fig). Although the discriminatory

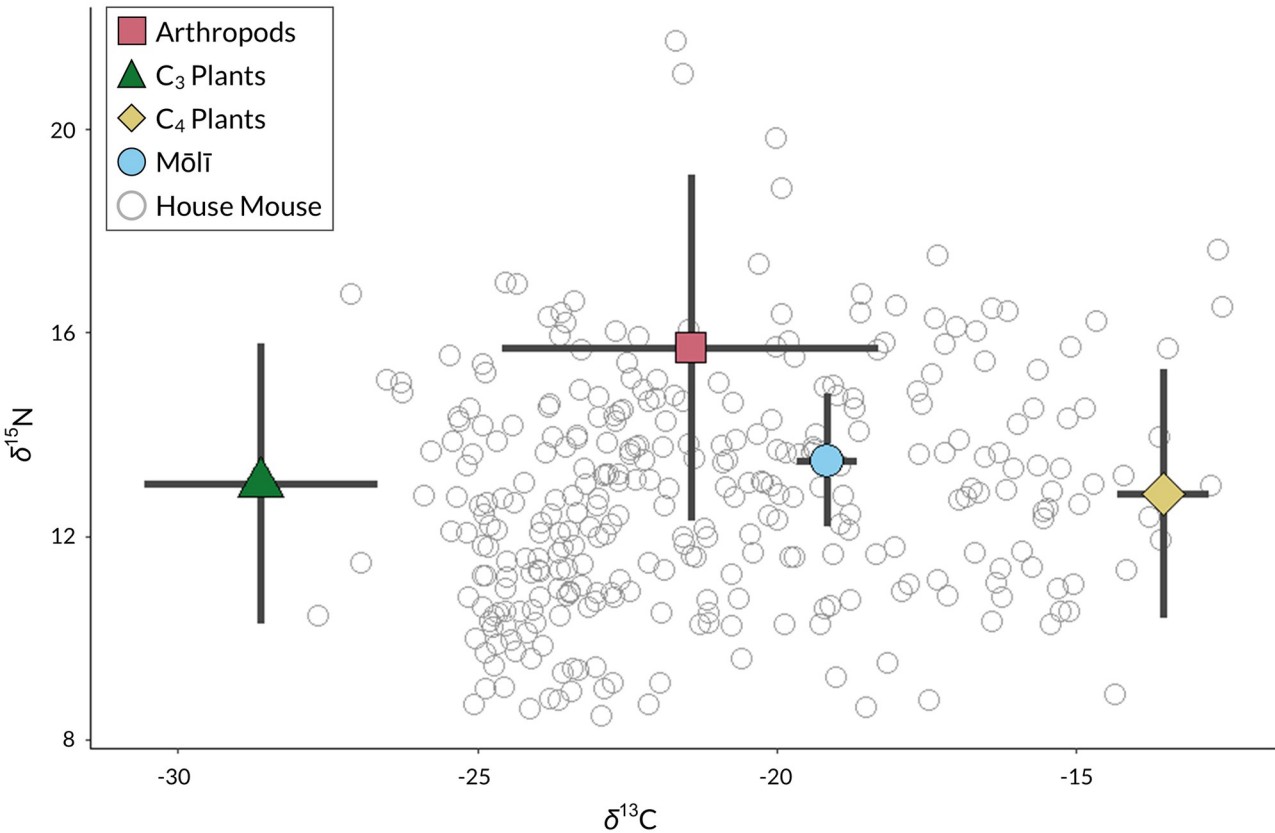

**Fig 2. $\delta^{13}$C and $\delta^{15}$N values of house mice and four main source groups.** Source groups include arthropods, C$_3$ plants, C$_4$ plants, and mōlī; mean values of $\delta^{13}$C and $\delta^{15}$N indicated by source group symbol and bars represent ± 1 SD. House mouse stable isotope values were adjusted by subtracting diet-tissue discrimination factors for hair tissue ($\delta^{13}$C = 1.7 ± 1.3‰; $\delta^{15}$N = 3.2 ± 1.1‰).

power of mixing models declines with seven or more prey sources [73], Bayesian mixing model frameworks such as MixSIAR can still estimate the posterior distributions of source proportions for under-determined mixing systems [42]. While we initially specified eight sources in our model, we combined sources *a posteriori* (*sensu* [42]) into four groups to summarize results: arthropods, C$_3$ plants, C$_4$ plants, and mōlī (Fig 2). Doing so preserves the covariation structure among source proportions while narrowing the distribution of possible solutions for each source [73]. In addition to our main stable isotope mixing model, we also ran an additional model excluding mōlī as a source, and *a posteriori* combined our sources into three groups: arthropods, C$_3$ plants, and C$_4$ plants (see S2 Appendix).

Because consumer tissues contain slightly elevated isotopic concentrations of C and N owing to discrimination during assimilation and excretion processes, it is necessary to correct isotopic data before analysis via a diet-tissue discrimination factor (DTDF; also commonly called a trophic discrimination factor [TDF] or trophic enrichment factor [TEF]; [76]). Before applying our mixing model, we evaluated several different DTDFs to determine the most appropriate fit given our consumer (mouse) and food taxa data sets; in other words, we sought to find the most suitable DTDF that would result in the highest probability that our proposed mixing models would explain the isotopic signature of Sand Island's mice. Some of the DTDFs we tested were specific to house mouse hair tissue, whereas other DTDFs were more broadly applicable to various taxa [77–84]. We used a Monte Carlo procedure from Smith et al. [85] to construct many possible mixing polygons for each DTDF to test if all consumer isotopic data

fell within the 95% mixing region. Based on this approach, we selected DTDF values estimated from the R package 'SIDER' [81] as the most appropriate fit; in the process, we also identified three outliers in our consumer data set (i.e., outside of the 95% mixing region) and excluded these data points from our mixing model (*sensu* [85]).

We then used the MixSIAR Bayesian mixing model framework [42] in R v4.1.2 [74] to quantify the proportion contributed to mouse diet by source group. We fitted our MixSIAR models with raw isotopic data of mouse hair samples, raw isotopic data of prey source groups (Fig 2 and S1 Fig), and the estimated DTDF for mouse hair ($\delta^{13}$C = 1.7 ± 1.3‰; $\delta^{15}$N = 3.2 ± 1.1‰). Because we had ASV data for all of our sources except for mōlī, we used "uninformative"/generalist Dirichlet priors for our mixing model as a conservative approach. We included habitat type and trapping session as random effects (S2 and S3 Figs) to assess the relative importance of these factors in explaining variation in source contribution via variance partitioning (see [42]). Additionally, we used a process*resid error structure to account for variability in consumer isotopic data [86] and incorporated elemental concentration dependence as well [87]. Our models consisted of three Markov Chain Monte Carlo (MCMC) chains of 1,000,000 iterations, thinned by 500, and with a burn-in of 500,000 [42]. We checked model convergence and fit by plotting the posterior predictive distributions and assessing Gelman-Rubi diagnostic values (i.e., Gelman-Rubin statistics < 1.1; *sensu* [19, 88]). *A posteriori*, we combined the original eight sources into four source groups for summarizing and reporting source proportion in diet; similarly for our model excluding mōlī as a source, we combined the original seven sources into three source groups.

## Results

### Overall diet composition

House mouse diet on Sand Island is dominated by arthropods (62%), with C$_3$ plants, C$_4$ plants, and mōlī contributing nearly equal proportions (15%, 12%, and 12%, respectively; Fig 3A, S4 Table). Specifically, Blattodea (cockroaches; *Periplaneta* sp.) are the main arthropod prey, comprising up to 23% of mouse diet (S5 Table). Araneae (stealthy ground spiders; *T. jaxartensis*) and Lepidoptera (moths) compose 13% and 14% respectively, while the two remaining arthropod sources—Diptera-Ixodida-Hymenoptera-Isopoda and the scuttle fly, *M. scalaris*—each constitute <10% of diet. When we excluded mōlī as a source group from our model, the contribution of arthropods to mouse diet increased to 73%, with C$_3$ and C$_4$ plants providing essentially the same proportion (14% and 13%, respectively; S2 Appendix).

**Arthropods.** We detected 56 genus-level arthropod ASVs from 14 orders and 41 families in mice, with a mean of 4.55 ASVs per mouse (SD = 3.07, range = 0–11). Most arthropod ASVs were rare, with 43 arthropod ASVs (77%) detected in <5% of samples (Table 1). However, several arthropod ASVs were commonly detected, with eight arthropod ASVs present in ≥30% of samples and 13 ASVs in ≥5% of samples. Nearly all arthropods consumed by mice are non-native. Of 13 commonly-detected arthropod ASVs, all are non-native species—except for *Ornithodoros capensis*, which has a worldwide distribution as a seabird ectoparasite. Additionally, of the 43 arthropod ASVs detected in <5% of samples, 39 ASVs (91%) are likely non-native; only two ASVs (dermestid beetles, *Dermestes ater* and *D. maculatus*) are native to the Hawaiian archipelago.

**Plants.** We detected 53 genus-level plant ASVs from 14 orders and 23 families in mice, with a mean of 2.71 ASVs per mouse (SD = 1.48, range = 0–9). Like arthropods, most plant ASVs were rare, with 42 ASVs (79%) detected in <5% of samples (Table 2). Eleven ASVs were detected in ≥5% of samples; only three plant ASVs occurred in ≥30% of samples. Most plants consumed by mice are non-native, and the two most frequently-occurring plant ASVs (in

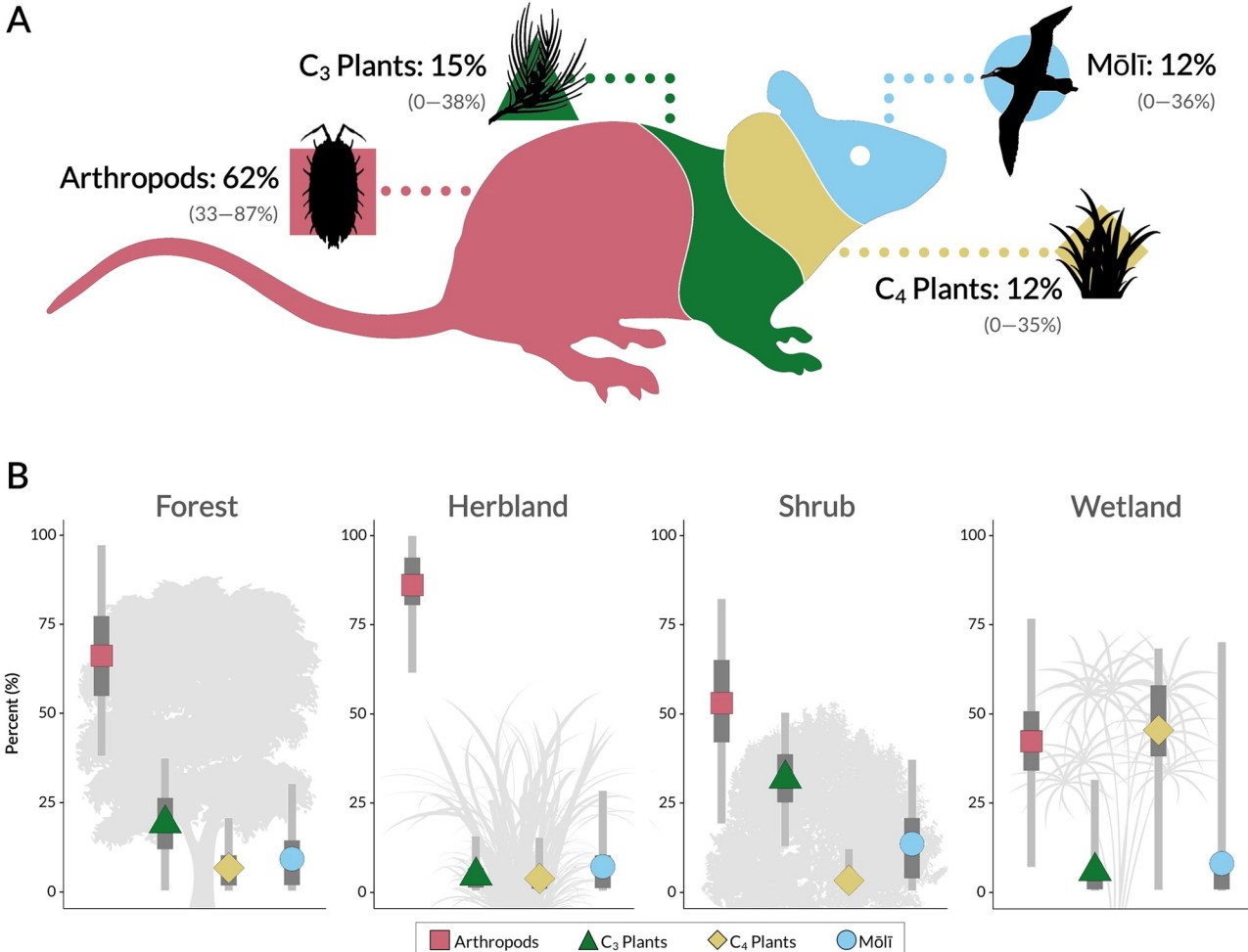

**Fig 3. Summary of stable isotope analyses for house mouse diet A) overall and B) by habitat type.** Results are reported as the dietary proportion of four (combined) food source groups: A) mean with 95% credible intervals; B) mean with 50% (thicker, dark gray bars) and 95% (thinner, light gray bars) credible intervals. Isopoda icon credit: Javier Luque (CC BY 3.0). All other icons are public domain or purchased with appropriate licenses via the Noun Project.

~50% of samples) are highly invasive: *Eleusine indica* and *Casuarina equisetifolia*. Of the remaining nine plant ASVs found in ≥5% of samples, four are non-native (*Lobularia maritima*, *Tournefortia argentea*, *Phyla nodiflora*, and *Stenotaphrum secundatum*) and four are native species found across Pacific Islands and/or within the Hawaiian archipelago (*Chenopodium album* [likely *C. oahuense* / 'āweoweo, present on Kuaihelani], *Ipomoea pes-caprae* / pōhuehue, *Boerhavia repens* / alena, and *Solanum nigrum* [closely related to *S. americanum* / pōpolo, present on Kuaihelani]). One ASV identified to genus has both native and non-native congeners present on Kuaihelani (*Eragrostis* sp. = *E. variabilis* / kāwelu [native], *E. paupera* [native], and *E. tenella* [non-native]). Of the 42 ASVs detected in <5% of samples, 32 ASVs (76%) are non-native, seven are native, and three have native and non-native congeners on Kuaihelani.

### Spatial and temporal variation in diet composition

**Habitat types.**    Arthropods are a core food source for mice across habitat types, composing 42–86% of mouse diet (Fig 3B, S4 Table). Specifically, *Periplaneta* sp. are the main

**Table 1. Arthropod ASVs detected in house mouse fecal samples through next-generation sequencing (NGS); *n* = 318 samples.**

| Order | Family | Genus | Species | Count | % Frequency of Occurrence | Status |
|---|---|---|---|---|---|---|
| Isopoda | Porcellionidae | *Porcellionides* | **Porcellionides pruinosus** | **199** | **62.6%** | Non-native |
| Isopoda | Porcellionidae | *Porcellio* | **Porcellio laevis** | **177** | **55.7%** | Non-native |
| Hymenoptera | Formicidae | *Monomorium* | **Monomorium bicolor complex BOLD:AAB9999[a]** | **150** | **47.2%** | Non-native |
| Hymenoptera | Formicidae | *Tetramorium* | **Tetramorium bicarinatum** | **144** | **45.3%** | Non-native |
| Diptera | Calliphoridae | *Lucilia* | **Lucilia sericata** | **127** | **39.9%** | Non-native |
| Diptera | Sarcophagidae | *Sarcophaga* | **Sarcophaga dux** | **116** | **36.5%** | Non-native |
| Ixodida | Argasidae | *Ornithodoros* | **Ornithodoros capensis** | **113** | **35.5%** | N/A (circumglobal distribution) |
| Blattodea | Blattidae | *Periplaneta* | **Periplaneta sp.[a]** | **104** | **32.7%** | Non-native |
| Hymenoptera | Formicidae | *Tetramorium* | **Tetramorium caldarium** | **61** | **19.2%** | Non-native |
| Diptera | Phoridae | *Megaselia* | **Megaselia scalaris[b]** | **40** | **12.6%** | Non-native |
| Araneae | Gnaphosidae | *Trachyzelotes* | **Trachyzelotes jaxartensis[b]** | **35** | **11.0%** | Non-native |
| Lepidoptera | Plutellidae | *Plutella* | **Plutella xylostella** | **24** | **7.5%** | Non-native |
| Lepidoptera | Tortricidae | *Olethreutes* | **Olethreutes sp.[b]** | **24** | **7.5%** | Non-native |
| Lepidoptera | Noctuidae | *Chrysodeixis* | *Chrysodeixis eriosoma* | 14 | 4.4% | Non-native |
| Lepidoptera | Tineidae | *Praeacedes* | *Praeacedes atomosella[b]* | 13 | 4.1% | Non-native |
| Lepidoptera | Saturniidae | *Archaeoattacus* | *Archaeoattacus edwardsii[b]* | 12 | 3.8% | Non-native |
| Isopoda | Armadillidae | *Cubaris* | *Cubaris murina[b]* | 12 | 3.8% | Non-native |
| Lepidoptera | Crambidae | *Hellula* | *Hellula undalis* | 7 | 2.2% | Non-native |
| Thysanoptera | Thripidae | *Arorathrips* | *Arorathrips mexicanus* | 6 | 1.9% | Non-native |
| Coleoptera | Dermestidae | *Dermestes* | *Dermestes ater* | 5 | 1.6% | Native |
| Diptera | Muscidae | Muscidae incertae sedis | Muscidae sp. BOLD:AAN8579[b] | 5 | 1.6% | Non-native |
| Hemiptera | Membracidae | *Vanduzea* | *Vanduzea sp. n. LA[a]* | 5 | 1.6% | Non-native |
| Coleoptera | Elateridae | *Heteroderes* | *Heteroderes kusuii[b]* | 4 | 1.3% | Non-native |
| Lepidoptera | Noctuidae | *Mythimna* | *Mythimna loreyi[a]* | 4 | 1.3% | Non-native |
| Psocoptera | Psocoptera incertae sedis | Psocoptera incertae sedis | Psocoptera sp. BOLD:ACB0983[b] | 4 | 1.3% | Non-native |
| Coleoptera | Curculionidae | *Sphenophorus* | *Sphenophorus venatus[a]* | 3 | 0.9% | Non-native |
| Coleoptera | Silvanidae | *Oryzaephilus* | *Oryzaephilus surinamensis* | 3 | 0.9% | Non-native |
| Diptera | Calliphoridae | *Chrysomya* | *Chrysomya megacephala* | 3 | 0.9% | Non-native |
| Lepidoptera | Autostichidae | *Stoeberhinus* | *Stoeberhinus testaceus* | 3 | 0.9% | Non-native |
| Thysanoptera | Thripidae | *Frankliniella* | *Frankliniella sp.[a]* | 3 | 0.9% | Non-native |
| Blattodea | Blattidae | *Periplaneta* | *Periplaneta americana* | 2 | 0.6% | Non-native |
| Thysanoptera | Phlaeothripidae | *Haplothrips* | *Haplothrips sp.[a]* | 2 | 0.6% | Non-native |
| Araneae | Nesticidae | *Eidmannella* | *Eidmannella sp. 1579[b]* | 1 | 0.3% | Non-native |
| Araneae | Salticidae | *Hasarius* | *Hasarius adansoni* | 1 | 0.3% | Non-native |
| Diplostraca | Macrotrichidae | *Macrothrix* | *Macrothrix sp. HE-364[b]* | 1 | 0.3% | Non-native |
| Blattodea | Rhinotermitidae | *Coptotermes* | *Coptotermes formosanus* | 1 | 0.3% | Non-native |
| Coleoptera | Anthribidae | *Araecerus* | *Araecerus fasciculatus* | 1 | 0.3% | Non-native |
| Coleoptera | Curculionidae | *Asynonychus* | *Asynonychus cervinus* | 1 | 0.3% | Non-native |
| Coleoptera | Dermestidae | *Dermestes* | *Dermestes maculatus* | 1 | 0.3% | Native |
| Diptera | Chironomidae | *Polypedilum* | *Polypedilum nubiferum* | 1 | 0.3% | Non-native |
| Diptera | Drosophilidae | Drosophilidae incertae sedis | Drosophilinae sp. BOLD:AAG8493[b] | 1 | 0.3% | Non-native |
| Diptera | Muscidae | *Thricops* | *Thricops cunctans[b]* | 1 | 0.3% | Non-native |
| Diptera | Sciaridae | Sciaridae incertae sedis | Sciaridae sp. JEF[b] | 1 | 0.3% | Non-native |

*(Continued)*

**Table 1.** (Continued)

| Order | Family | Genus | Species | Count | % Frequency of Occurrence | Status |
|-------|--------|-------|---------|-------|---------------------------|--------|
| Hemiptera | Cicadellidae | Cicadellidae incertae sedis | Cicadellidae sp. 3 AY-2015[b] | 1 | 0.3% | Non-native |
| Hemiptera | Cicadellidae | *Neoaliturus* | *Neoaliturus tenellus*[b] | 1 | 0.3% | Non-native |
| Hemiptera | Coccidae | *Coccus* | *Coccus viridis* | 1 | 0.3% | Non-native |
| Hemiptera | Delphacidae | *Toya* | *Toya* sp. BOLD:AAF8686[a] | 1 | 0.3% | Non-native |
| Hymenoptera | Braconidae | *Chelonus* | *Chelonus blackburni* | 1 | 0.3% | Non-native |
| Hymenoptera | Formicidae | *Plagiolepis* | *Plagiolepis alluaudi* | 1 | 0.3% | Non-native |
| Hymenoptera | Vespidae | *Polistes* | *Polistes jokahamae* | 1 | 0.3% | Non-native |
| Lepidoptera | Noctuidae | *Spodoptera* | *Spodoptera mauritia* | 1 | 0.3% | Non-native |
| Lepidoptera | Tineidae | Tineidae incertae sedis | Tineidae sp. PTO-847.19[b] | 1 | 0.3% | Non-native |
| Psocoptera | Liposcelidae | *Liposcelis* | *Liposcelis decolor*[b] | 1 | 0.3% | Non-native |
| Psocoptera | Liposcelidae | *Liposcelis* | *Liposcelis entomophila*[b] | 1 | 0.3% | Non-native |
| Decapoda | Decapoda incertae sedis | Decapoda incertae sedis | Brachyura sp. LPdivOTU430 | 1 | 0.3% | N/A; not enough data |
| Cyclopoida | Cyclopoida incertae sedis | Cyclopoida incertae sedis | Cyclopoida sp. HE-372.1 | 1 | 0.3% | N/A; not enough data |

ASVs in bold were detected in ≥5% of fecal samples. Status column based on the classification scheme from [89, 90].

[a]Indicates an ASV for which native and non-native congeners exist on Kuaihelani.

[b]Indicates an ASV that has not been observed on Kuaihelani, nor has any congeners present, and would be considered non-native.

arthropod source in mouse diet in all habitats, except the wetland; in the herbland, in particular, the majority of mouse diet comes from *Periplaneta* sp. and *T. jaxartensis* (S5 Table). $C_3$ and $C_4$ plant contributions vary substantially among mice (Fig 3B); $C_3$ and $C_4$ plants are minimally consumed in the herbland, whereas $C_3$ plants are an important source in both shrub and forest habitats (31% and 19% respectively), and $C_4$ plants contribute the greatest proportion to mouse diet in wetlands (45%) (S4 and S5 Tables). Mōlī contribution to mouse diet is relatively low (between 7–13%)—but consistent—among all habitat types (Fig 3B), with a slightly elevated proportion in shrub habitat (S4 and S5 Tables). Similar results were found across habitat types with the model excluding mōlī as a source group (S2 Appendix).

Multivariate analyses reveal that mice consume significantly different compositions of plants and arthropods among habitats (Table 3). Habitat type has a stronger effect on plant diet, in particular; mice are 36% dissimilar in their plant consumption among habitats, but only 9% dissimilar in arthropod consumption (see "Component" [components of variation estimates] in Table 3). Indeed, habitat effects are visibly evident in plant diet composition (Fig 4A). In Fig 4A, mice cluster into loose groups that largely correspond to habitat types; this is especially apparent in contrasts of mice from the herbland with those from the forest across PCoA axis 1. This difference in plant diet composition is also evident in ASV data (S4 Fig). For example, over 90% of mice captured in the forest consumed *Casuarina equisetifolia*, compared to only 28%-38% of mice from other habitats (S4 Fig). Differences in arthropod consumption by mice, however, are relatively weak among habitat types (i.e., mice overlap in diet among habitats; Fig 4A). Although variability in plant composition of mouse diet differs among habitats, it does not for arthropod composition (Table 3, S5 Fig); pairwise comparisons further support these findings (S6 and S7 Tables). In summary, plant consumption by mice differs significantly in terms of composition and variability thereof among habitat types, whereas arthropod consumption differs (weakly) only among habitats. Notably, most of the variation

**Table 2. Plant ASVs detected in house mouse fecal samples through next-generation sequencing (NGS); *n* = 318 samples.**

| Order | Family | Genus | Species | Count | % Frequency of Occurrence | Status |
|---|---|---|---|---|---|---|
| Poales | Poaceae | *Eleusine* | ***Eleusine indica*** | **160** | **50.3%** | Non-native; invasive |
| Fagales | Casuarinaceae | *Casuarina* | ***Casuarina equisetifolia*** | **155** | **48.7%** | Non-native; invasive |
| Caryophyllales | Chenopodiaceae | *Chenopodium* | ***Chenopodium album*** | **100** | **31.4%** | Native |
| Brassicales | Brassicaceae | *Lobularia* | ***Lobularia maritima*** | **59** | **18.6%** | Non-native; naturalized |
| Boraginales | Heliotropiaceae | *Tournefortia* | ***Tournefortia argentea*** | **53** | **16.7%** | Non-native; naturalized |
| Solanales | Convolvulaceae | *Ipomoea* | ***Ipomoea pes-caprae*** | **45** | **14.2%** | Native |
| Caryophyllales | Nyctaginaceae | *Boerhavia* | ***Boerhavia repens*** | **43** | **13.5%** | Native |
| Lamiales | Verbenaceae | *Phyla* | ***Phyla nodiflora*** | **38** | **11.9%** | Non-native; naturalized |
| Poales | Poaceae | *Eragrostis* | ***Eragrostis* sp.[a]** | **20** | **6.3%** | N/A |
| Solanales | Solanaceae | *Solanum* | ***Solanum nigrum*** | **20** | **6.3%** | Native |
| Poales | Poaceae | *Stenotaphrum* | ***Stenotaphrum secundatum*** | **18** | **5.7%** | Non-native; naturalized |
| Asterales | Goodeniaceae | *Scaevola* | *Scaevola taccada* | 15 | 4.7% | Native |
| Caryophyllales | Aizoaceae | *Sesuvium* | *Sesuvium portulacastrum* | 14 | 4.4% | Native |
| Caryophyllales | Caryophyllaceae | *Spergularia* | *Spergularia* sp.[b] | 11 | 3.5% | Non-native |
| Poales | Poaceae | *Cynodon* | *Cynodon aristulatus*[b] | 10 | 3.1% | Non-native |
| Boraginales | Heliotropiaceae | *Euploca* | *Euploca campestris*[b] | 8 | 2.5% | Non-native |
| Poales | Poaceae | *Eragrostis* | *Eragrostis tenella* | 7 | 2.2% | Non-native; naturalized |
| Brassicales | Brassicaceae | *Lepidium* | *Lepidium angustissimum*[b] | 7 | 2.2% | Non-native |
| Asterales | Asteraceae | *Conyza* | *Conyza* sp. | 6 | 1.9% | Non-native; invasive |
| Malpighiales | Euphorbiaceae | *Euphorbia* | *Euphorbia* sp. | 6 | 1.9% | Non-native; naturalized |
| Poales | Poaceae | *Cynodon* | *Cynodon* sp.[b] | 5 | 1.6% | Non-native |
| Poales | Cyperaceae | *Cyperus* | *Cyperus polystachyos* | 4 | 1.3% | Native |
| Poales | Poaceae | *Sporobolus* | *Sporobolus pyramidatus* | 4 | 1.3% | Non-native; naturalized |
| Asterales | Asteraceae | *Erigeron* | *Erigeron canadensis* | 4 | 1.3% | Non-native; invasive |
| Caryophyllales | Caryophyllaceae | *Spergularia* | *Spergularia media*[b] | 4 | 1.3% | Non-native |
| Myrtales | Combretaceae | *Terminalia* | *Terminalia catappa* | 4 | 1.3% | Non-native; naturalized |
| Solanales | Convolvulaceae | *Ipomoea* | *Ipomoea indica* | 4 | 1.3% | Native |
| Boraginales | Heliotropiaceae | *Euploca* | *Euploca* sp.[b] | 3 | 0.9% | Non-native |
| Poales | Poaceae | *Cynodon* | *Cynodon nlemfuensis*[b] | 2 | 0.6% | Non-native |
| Poales | Poaceae | *Dactyloctenium* | *Dactyloctenium aegyptium* | 2 | 0.6% | Non-native; invasive |
| Poales | Poaceae | *Eustachys* | *Eustachys* sp.[b] | 2 | 0.6% | Non-native |
| Asterales | Asteraceae | *Pseudognaphalium* | *Pseudognaphalium* sp. | 2 | 0.6% | Native |
| Brassicales | Brassicaceae | *Lepidium* | *Lepidium virginicum* | 2 | 0.6% | Non-native; naturalized |
| Caryophyllales | Amaranthaceae | *Amaranthus* | *Amaranthus spinosus* | 2 | 0.6% | Non-native; invasive |
| Caryophyllales | Amaranthaceae | *Amaranthus* | *Amaranthus viridis* | 2 | 0.6% | Non-native; naturalized |
| Malpighiales | Euphorbiaceae | *Euphorbia* | *Euphorbia peplus* | 2 | 0.6% | Non-native; naturalized |
| Rosales | Urticaceae | *Urtica* | *Urtica dioica*[c] | 2 | 0.6% | Non-native |
| Poales | Cyperaceae | *Cyperus* | *Cyperus* sp.[a] | 1 | 0.3% | N/A |
| Poales | Poaceae | *Digitaria* | *Digitaria* sp.[b] | 1 | 0.3% | Non-native |
| Poales | Poaceae | *Ectrosia* | *Ectrosia* sp.[c] | 1 | 0.3% | Non-native |
| Poales | Poaceae | *Hordeum* | *Hordeum* sp.[b] | 1 | 0.3% | Non-native |
| Poales | Poaceae | *Poa* | *Poa annua* | 1 | 0.3% | Non-native; naturalized |
| Poales | Poaceae | *Sporobolus* | *Sporobolus virginicus* | 1 | 0.3% | Native |
| Apiales | Apiaceae | *Cyclospermum* | *Cyclospermum leptophyllum* | 1 | 0.3% | Non-native; naturalized |
| Asterales | Asteraceae | *Bidens* | *Bidens alba* | 1 | 0.3% | Non-native; invasive |
| Caryophyllales | Chenopodiaceae | *Atriplex* | *Atriplex suberecta* | 1 | 0.3% | Non-native; naturalized |
| Fabales | Fabaceae | *Glycine* | *Glycine max* | 1 | 0.3% | Non-native; naturalized |

*(Continued)*

**Table 2.** (Continued)

| Order | Family | Genus | Species | Count | % Frequency of Occurrence | Status |
|---|---|---|---|---|---|---|
| Myrtales | Onagraceae | *Oenothera* | *Oenothera* sp.[b] | 1 | 0.3% | Non-native |
| Rosales | Cannabaceae | *Humulus* | *Humulus* sp.[c] | 1 | 0.3% | Non-native |
| Solanales | Convolvulaceae | *Ipomoea* | *Ipomoea* sp.[a] | 1 | 0.3% | N/A |
| Solanales | Convolvulaceae | *Ipomoea* | *Ipomoea nil*[b] | 1 | 0.3% | Non-native |
| Solanales | Solanaceae | *Solanum* | *Solanum* sp.[a] | 1 | 0.3% | N/A |
| Zygophyllales | Zygophyllaceae | *Tribulus* | *Tribulus terrestris* | 1 | 0.3% | Native |

ASVs in bold were detected in ≥5% of fecal samples. Status column based on weed classification scheme from [91].

[a]Indicates an ASV for which native and non-native congeners exist on Kuaihelani.

[b]Indicates an ASV that has not been observed on Kuaihelani, but its status is based on those of congeners present on Kuaihelani.

[c]Indicates an ASV that has not been observed on Kuaihelani, nor has any congeners present, and would be considered non-native.

in plant and arthropod diet composition is at the individual level; here, mice are 42–49% dissimilar in their plant and arthropod consumption, respectively (see "Component" in Table 3).

**Trapping session.** Plant and arthropod composition of mouse diet differs significantly among trapping sessions (Table 3). However, this seasonal effect was not clearly apparent and much overlap occurred in both plant and arthropod diet composition across sessions (Fig 4B, Table 3). While variability in plant composition of mouse diet differs significantly among trapping sessions, variability in arthropod composition does not (PERMDISP results in Table 3, S6 and S7 Tables). Compared to variation in diet owing to habitat type, mice captured during different sessions are only 17% dissimilar in their plant diet composition and hardly vary in arthropod diet composition (only ~5%; see "Component" in Table 3, S6 and S7 Figs).

**Table 3. PERMANOVA partitioning and analysis and PERMDISP test results of plant ASVs (53 taxa) and arthropod ASVs (56 taxa) detected in house mouse fecal samples, based on Sørensen dissimilarities.**

**Plant Diet Composition**

| Source | | | PERMANOVA | | | | | | PERMDISP | | | |
|---|---|---|---|---|---|---|---|---|---|---|---|---|
| Source | df | *SS* | MS | Pseudo *F* | *p* | Component | Var | SD | $df_1$ | $df_2$ | *F* | *p* |
| Habitat Type | 3 | 285320 | 95108 | 53.13 | < **0.001** | Fixed | 1291.40 | 35.94 | 3 | 308 | 24.54 | < **0.001** |
| Trapping Session | 6 | 87096 | 14516 | 8.11 | < **0.001** | Fixed | 299.37 | 17.30 | 6 | 305 | 12.00 | < **0.001** |
| Habitat Type x Trapping Session | 18 | 84027 | 4668 | 2.61 | < **0.001** | Fixed | 264.29 | 16.26 | 27 | 284 | 2.66 | < **0.001** |
| Residual | 284 | 508350 | 1790 | - | - | Random | 1790.00 | 42.31 | | | | |
| Total | 311 | 971680 | - | - | - | | - | - | | | | |

**Arthropod Diet Composition**

| Source | | | PERMANOVA | | | | | | PERMDISP | | | |
|---|---|---|---|---|---|---|---|---|---|---|---|---|
| Source | df | *SS* | MS | Pseudo *F* | *p* | Component | Var | SD | $df_1$ | $df_2$ | *F* | *p* |
| Habitat Type | 3 | 24045 | 8015 | 3.34 | < **0.001** | Fixed | 81.98 | 9.05 | 3 | 289 | 2.72 | 0.133 |
| Trapping Session | 6 | 21933 | 3656 | 1.52 | **0.023** | Fixed | 31.27 | 5.59 | 6 | 286 | 1.16 | 0.567 |
| Habitat Type x Trapping Session | 18 | 49001 | 2722 | 1.14 | 0.167 | Fixed | 31.66 | 5.63 | 27 | 265 | 1.20 | 0.753 |
| Residual | 265 | 635410 | 2398 | - | - | Random | 2397.80 | 48.97 | | | | |
| Total | 292 | 735080 | - | - | - | | - | - | | | | |

Pseudo *F* statistics were calculated for each effect via direct analogs to univariate expectations of mean squares (EMS). Each effect is identified as contributing either a fixed or random component to the overall model; "Var" gives the estimated sizes of the components of variation, based on multivariate analogs to classical ANOVA unbiased estimators; "SD" provides the square root of these values and is in Sørensen units.

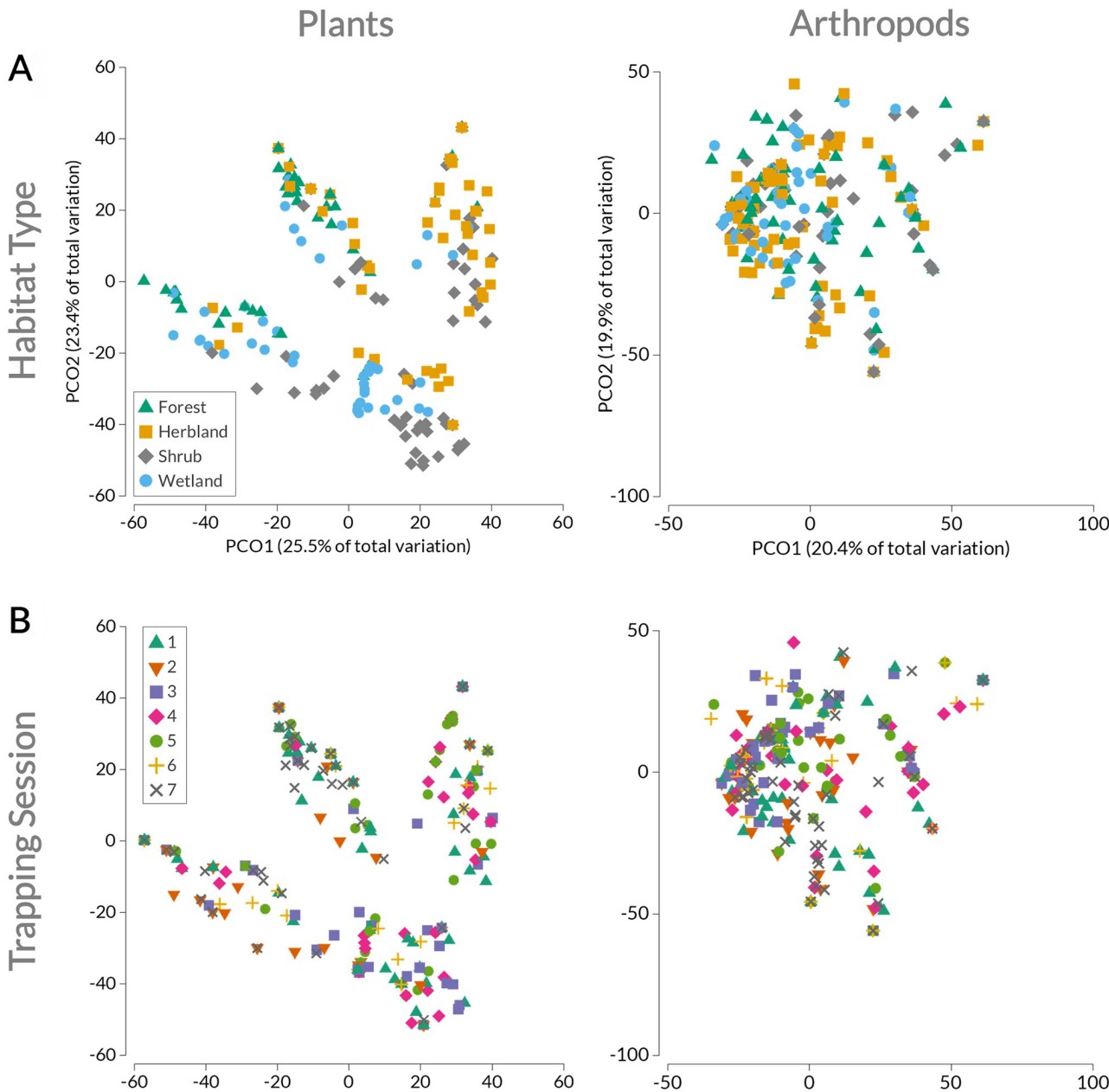

**Fig 4. PCoA of plant and arthropod consumption by house mice (*n* = 318) based on Sørensen dissimilarities by A) habitat type and B) trapping session.** Points located closer together represent more similar groups of plants or arthropods consumed by mice, and points farther away represent less similar groups.

Although we cannot reliably interpret source contribution among trapping sessions from our stable isotope mixing model, all four source contributions remain largely even among sessions (S4 Table). Based on the estimated variation in diet for habitat type and trapping session in our model, habitat contributes nearly four times more to diet variation than does trapping session, which also mirrors our multivariate statistical findings.

**Habitat type and trapping session interaction.** We detected an interaction between habitat type and trapping session for plant consumption only, but this interaction was very weak

compared to these two factors independently. This effect is likely driven by differences in plant consumption (and/or variability thereof) by mice among habitats for each trapping session, rather than a temporal effect within each habitat (S6 Table).

## Discussion

By combining next-generation sequencing (NGS) and stable isotope analysis (SIA), we uncovered accurate, high-resolution information about which taxa house mice consume on Sand Island and to what extent. Overall, we found that arthropods are the main food source, followed by nearly equal contributions of $C_3$ plants, $C_4$ plants, and mōlī. Most taxa identified in mouse diet are non-native, which may be released post-eradication and hinder recovery of Sand Island's ecosystem. Altogether, our study demonstrates the importance and application of mouse diet to predict and prepare for eradication outcomes, and it adds to the growing literature on the ecological effects of invasive house mice on islands.

### High occurrence of non-native taxa in diet

Sand Island's mice predominantly consume non-native arthropods and plants—a diet that strongly mirrors resources available in this ecosystem. Kuaihelani has undergone drastic landscape changes in the last century, coupled with a complex history of human-facilitated species introductions [45]. For example, of over 400 plant species observed across Kuaihelani since 1902, >90% are non-native [91]. Similarly, of the ~600 arthropod species documented since 1894, upwards of 80% are non-native [89, 90].

In such a non-native dominated ecosystem, it is unsurprising to find mouse diet dominated by non-natives. Indeed, other studies from island systems with a history of disturbance and introduced species (such as the Main Hawaiian Islands and Southeast Farallon Island) also report that mice consume a mix of native and non-native taxa [17, 19, 92]. As opportunistic omnivores, mice will likely eat accessible and abundant taxa [16]. However, no other house mouse diet study has recorded such a high occurrence of non-native taxa.

The prevalence of non-native taxa in mouse diet contrasts with dietary studies from other island systems, which are arguably more "pristine" (or less heavily managed) than Kuaihelani. For example, many subantarctic islands are very isolated and support high levels of endemism with few introduced species [6, 18, 21]. In turn, consumption of non-native taxa by invasive mice appears low across these systems [6]—but few studies have explicitly mentioned or discussed this aspect of mouse diet or mouse preference for native versus non-native dietary items. Instead, most studies focus on elucidating which native species mice consume and the subsequent ecological ramifications while overlooking non-native contributions to diet [16, 18, 21, 93].

### Contribution of detrital-feeding arthropods

Generally, mice on islands target large-bodied arthropods, such as Coleoptera, Orthoptera, and Lepidopteran larvae [9, 13, 14, 92]. Instead, Sand Island mice consume several detrital-feeding and necrophagous arthropods, including isopods, flesh flies, ants, and cockroaches, as well as a seabird ectoparasite, *Ornithodoros capensis* (Table 1, S3 Table). For example, three dipteran species common in Sand Island mouse diet (*Lucilia sericata*, *Sarcophaga dux*, and *Megaselia scalaris*) specifically rely on carrion during their life cycle, laying their eggs and larvae in carcasses. With >3 million birds nesting on or migrating through Kuaihelani annually, carcasses are widely available and likely are an important source for dipteran reproduction. Undoubtedly, mice scavenge decaying seabird tissue, simultaneously eating detrital-feeding arthropods (and their eggs and larvae) in and around seabird carcasses.

Consumption of detritophagous and necrophagous arthropods also explains certain SIA results, as arthropods appear to be $\delta^{15}$N-enriched compared to mōlī (Fig 2). As a whole, Kuaihelani is $\delta^{15}$N-enriched owing to high nitrogen input and deposition via seabird chick and adult carcasses, eggs, guano, and seabird regurgitate/boluses, like many other seabird islands [94]. Therefore, arthropods feeding on seabird tissues and by-products become $\delta^{15}$N-enriched, which is also evident in mice that eat these arthropods (Fig 2, S1 Fig).

## Sources of variation in diet

Mice tend to consume the most common taxa in their surrounding environment, which is especially apparent with plants (Fig 4, Table 3, S4 and S5 Tables). For example, two species that commonly occur along the beach strand—*Tournefortia argentea* and *Ipomoea pes-caprae* —were detected in >50% of mice in the shoreline shrub habitat (and composed nearly a third of mouse diet as $C_3$ plants; Fig 3B) but were rarely found in mice from habitats further inland (S4 Fig). This trend, however, is less evident for arthropods, as mice consume largely the same array of arthropods across habitats (Fig 4, Table 3, S5 Fig). This finding complements previous research on arthropod community composition and structure across Kuaihelani [63], in that arthropod communities share similar composition and structure across habitats. Isotopically though, it is clear that mice rely on sources differently among habitats, which is likely driven by differences in plant communities and small home ranges [95].

Sand Island's mice have relatively weak temporal variation in their diet compared to that of mice from other islands [13, 19, 21, 88]. Seasonal differences in mouse consumption of plants are likely tied to pronounced dry/wet seasons on Kuaihelani and associated availability of plant sources. Arthropod consumption, however, varies minimally throughout the year and corresponds to temporal patterns of arthropod abundance across Kuaihelani [63]. Namely, arthropods consumed by mice on Sand Island (Isopoda, Diptera, and Acari) have comparable abundances on Eastern Island (which lacks mice; [63]). Thus, sufficient arthropod sources exist on Sand Island that are seemingly unaffected by mouse predation. This may explain the dominance of arthropods in mouse diet; because they are abundant and accessible year-round, arthropods may be eaten by mice owing to preference and not because of seasonal availability (see [14]). Although our single year of sampling is not indicative of weather extremes and other environmental variation that have been documented on Kuaihelani, our study does provide valuable information for eradication planning and post-eradication monitoring.

## Limitations

Every method used for diet analysis has benefits and drawbacks. Similarly, next-generation sequencing (NGS) and stable isotope analysis (SIA) have weaknesses that may bias or limit inferences. NGS alone cannot be used to assess the contribution of different taxa via sequence read counts as it provides only relative abundance of amplicon targets. In addition, these approaches are limited by the target range of the primers and biases of PCR amplification (e.g., [96]). In particular, it may be difficult to detect and amplify the DNA of prey taxa, especially in highly degraded tissues such as fecal samples [41, 97]. Other studies have also encountered problems in detecting avian DNA in mammalian feces and stomach samples (see [32]). Moreover, soft tissue (such as epithelial or muscle tissue, or blood) may be more easily digestible and assimilated than harder substances (i.e., plant fibers and chitin) and therefore be less abundant and/or of a lower quality in fecal samples (see [98]). Thus, although sequence data are high-resolution at the taxonomic level, they provide only a semi-quantitative description of diet in terms of the presence/absence of food taxa among samples [66].

Camera footage and photos have clearly documented mice attacking and feeding on Sand Island's mōlī [4, 10]—but, we were unable detect seabird DNA in mouse fecal samples. The inability to detect DNA, however, does not mean mice do not consume seabird tissue (see S1 Appendix). Rather, we lack tools that adequately amplify and differentiate degraded seabird DNA from high-quality host (mouse) DNA. We encourage future research on developing primer sets of different genes and genetic regions to better detect seabird DNA in rodent fecal samples. In the meantime, other methods such as camera traps and behavioral studies must be used to clarify rodent consumption of seabirds (see [26, 27]).

SIA is a common method to quantify source contribution to diet, but it may produce more diffuse estimates for omnivorous consumers that rely on a diverse array of sources or when sources are isotopically similar (*sensu* [73]). For Sand Island's mice and their source groups, we found weak negative correlations (ranging from r = -0.07 to -0.27) between the eight main source groups, indicating that our model could distinguish and estimate the contribution of these sources. Although the range of possible diet proportions narrowed when we combined sources *a posteriori* into four source groups, the 95% credible intervals were still relatively wide (S4 and S5 Tables, S2 Appendix). This is unsurprising, given the context of Kuaihelani as a seabird island ecosystem; seabirds deposit nutrients (especially N), which are consumed or broken down by arthropods and also absorbed by plants. In turn, mice consume detrital-feeding and necrophagous arthropods (which also consume seabird carcasses) and other taxa that are all isotopically-enriched from the same seabird-derived nutrients—hence, dietary proportion estimates are somewhat broad in this study (as can be expected in other seabird island systems, too). Also, because native and non-native plants were not isotopically distinct (other than broadly splitting into $C_3$ and $C_4$ plant groups), we were unable to estimate dietary contributions from native and non-native sources.

## Implications for management

Eradications of invasive rodents are effective island restoration tools and especially benefit threatened bird populations by removing their predators [99, 100]. In addition, it is assumed that other native biota will recover quickly in the absence of predatory and competitive effects. However, non-native and invasive taxa are also released once rodents are removed, which can cause adverse, landscape-wide changes—but few studies have examined this phenomenon. On Palmyra Atoll NWR, eradication of black rats (*Rattus rattus*) resulted in the recovery of native trees [101]. On the other hand, introduced *Cocos nucifera* also proliferated quickly, owing to the absence of seed predation by rats [101, 102]. Despite the initial recovery of native biota, the release of a non-native species that does not provide high-quality seabird nesting habitat had the potential to undermine the efficacy of this rat eradication [102]. Control efforts for *C. nucifera* are in progress to re-establish the atoll's native forests—a cost that could have been reduced if conducted pre-emptively (now, upwards of $2.2 million; [103]).

Like the ecosystem response observed on Palmyra Atoll, we predict that non-native taxa will be released following mouse eradication on Sand Island. Without proactive, pre-eradication control efforts, invasive species consumed by mice (such as plants—*Casuarina equisetifolia* [104] and *Eleusine indica* [105]) will likely increase in abundance and distribution after mouse removal. Native plants may initially increase in recruitment and cover, but non-native plants previously suppressed by mice may outcompete native plants and spread further on Sand Island, resulting in an "invasional release" (also called the "Sisyphus effect"; [106]). The release of non-native and invasive plants could pose especially serious and costly challenges to ongoing conservation and restoration efforts on Sand Island. For instance, >$2 million has

been spent controlling only *one* invasive plant species (*Verbesina encelioides*) across Kuaihelani [107]. Although mice do not consume *V. encelioides*, they do eat several other highly invasive plants, which may irrupt post-eradication and could require additional time, energy, and funding to control. Similarly, we predict that arthropod community composition on Sand Island will change substantially (because arthropods compose the majority of mouse diet), which may affect ecosystem processes such as nutrient cycling as well as food availability for terrestrial birds, migratory shorebirds, and larger arthropods.

Connecting predictions to outcomes of how highly-modified island ecosystems respond to rodent removal constitutes a widespread problem in invasion biology and restoration ecology. In the case of Sand Island, invasive house mice consume *both* native and non-native taxa. While native taxa should respond favorably to mouse removal (e.g., [99, 100]), the response trajectories for non-native taxa are ambiguous. Changing dynamics in the composition and structure of biotic communities following mouse eradication have serious implications for current and future restoration efforts. More importantly, disturbed systems—such as Sand Island —may not realign to native-dominated landscapes, even with the eradication of invasive predators. Highly-altered systems can resist restoration efforts and remain in a deteriorated, alternative state [108] despite ongoing, intensive management.

Given that Sand Island has undergone extensive landscape modification and has a long history of species introductions, is it worth eradicating mice from this island system? Mouse eradication will undeniably benefit the millions of birds that use Kuaihelani as a safe haven for breeding and during migration; in turn, the conservation of robust seabird colonies will ensure vital nutrient transfer between land and sea [109]. However, we emphasize that for highly-managed (or "disturbed") islands, implementing pre- and post-eradication management efforts are key to mitigating "surprise effects" or adverse outcomes [110]. Eradication projects should take the relationships of the target species and the broader ecosystem into account [111]. Naturally, eradication operations will differ from island to island and will require a place-based approach [112] based on conservation objectives and the context (i.e., history and anthropogenic modifications) of the island system. However, with more eradications planned on larger, increasingly complex islands with many non-native species, we face a critical knowledge gap of ecosystem restoration outcomes (and challenges) following rodent removal. Here, Sand Island is an important case study to understand interactions of invasive species on islands and how they influence ecosystems.

Conservation practitioners should use diet analysis as a preliminary measure to understand trophic interactions of target species and predict how affected taxa may respond following eradication. Diet analysis can serve as a framework to prioritize pre- and post-eradication ecological monitoring by measuring the response (e.g., demography) of taxa most directly impacted by rodents (as prey). The status of food taxa (i.e., native, non-native, invasive) is especially important for setting realistic restoration objectives; impact assessment protocols can help to prioritize which species may pose the most severe threats post-eradication [113]. Such knowledge can inform pre-eradication management to avoid unexpected irruptions of previously-suppressed species, which may set back ongoing restoration and recovery efforts of native insular biodiversity [114, 115].

## Supporting information

**S1 Fig. $\delta^{13}$C and $\delta^{15}$N values of house mice and eight main source groups.** Source groups include $C_3$ plants, $C_4$ plants, Diptera-Ixodida-Hymenoptera-Isopoda (DIP-IXO-HYM-ISO), Lepidoptera, *Megaselia scalaris*, mōlī, *Periplaneta* sp., and *Trachyzelotes jaxartensis;* mean values of $\delta^{13}$C and $\delta^{15}$N indicated by source group symbol and bars represent ± 1 SD. House

mouse stable isotope values were adjusted by subtracting diet-tissue discrimination factors for hair tissue ($\delta^{13}$C = 1.7 ± 1.3‰; $\delta^{15}$N = 3.2 ± 1.1‰).
(TIF)

**S2 Fig. $\delta^{13}$C and $\delta^{15}$N values of house mice and four (combined) source groups for each habitat type.** Source groups include arthropods, C$_3$ plants, C$_4$ plants, and mōlī; mean values of $\delta^{13}$C and $\delta^{15}$N indicated by source group symbol and bars represent ± 1 SD. House mouse stable isotope values were adjusted by subtracting diet-tissue discrimination factors for hair tissue ($\delta^{13}$C = 1.7 ± 1.3‰; $\delta^{15}$N = 3.2 ± 1.1‰).
(TIF)

**S3 Fig. $\delta^{13}$C and $\delta^{15}$N values of house mice and four (combined) source groups for each trapping session.** Source groups include arthropods, C$_3$ plants, C$_4$ plants, and mōlī; mean values of $\delta^{13}$C and $\delta^{15}$N indicated by source group symbol and bars represent ± 1 SD. House mouse stable isotope values were adjusted by subtracting diet-tissue discrimination factors for hair tissue ($\delta^{13}$C = 1.7 ± 1.3‰; $\delta^{15}$N = 3.2 ± 1.1‰).
(TIF)

**S4 Fig. Percent frequency of occurrence (%FOO) of plant ASVs detected in ≥5% of house mouse fecal samples across habitat types.** [a]Indicates an ASV for which native and non-native congeners exist on Kuaihelani.
(TIF)

**S5 Fig. Percent frequency of occurrence (%FOO) of arthropod ASVs detected in ≥5% of house mouse fecal samples across habitat types.** [a]Indicates an ASV for which native and non-native congeners exist on Kuaihelani. [b]Indicates an ASV that has not been observed on Kuaihelani, nor has any congeners present, and would be considered non-native.
(TIF)

**S6 Fig. Percent frequency of occurrence (%FOO) of plant ASVs detected in ≥5% of house mouse fecal samples across trapping sessions.** [a]Indicates an ASV for which native and non-native congeners exist on Kuaihelani.
(TIF)

**S7 Fig. Percent frequency of occurrence (%FOO) of arthropod ASVs detected in ≥5% of house mouse fecal samples across trapping sessions.** [a]Indicates an ASV for which native and non-native congeners exist on Kuaihelani. [b]Indicates an ASV that has not been observed on Kuaihelani, nor has any congeners present, and would be considered non-native.
(TIF)

**S1 Table. Description of Sand Island habitat types.**
(XLSX)

**S2 Table. Summary of house mice captured among traplines on Sand Island from April 2018 to May 2019.** A) Count summary of all house mice (sampling pool) captured from April 2018 to May 2019 (by sex, age, habitat type, and trapping session). B) Count summary of house mice used for next-generation sequencing and stable isotope analysis (by sex, habitat type, and trapping session).
(XLSX)

**S3 Table. Description of feeding ecology for common arthropod taxa detected in house mouse diet.**
(XLSX)

**S4 Table. Summary of the estimated proportions from four (combined) source groups in house mouse diet overall and among habitat types and trapping sessions.** Source proportions are reported overall and for each habitat type and trapping session with mean values, standard deviation, and 95% credible intervals.
(XLSX)

**S5 Table. Summary of the estimated proportions from eight source groups in house mouse diet overall and among habitat types and trapping sessions.** Source proportions are reported overall and for each habitat type and trapping session with mean values, standard deviation, and 95% credible intervals.
(XLSX)

**S6 Table. Pairwise PERMANOVA and PERMDISP test results for plant ASVs (53 taxa) detected in house mouse fecal samples, based on Sørensen dissimilarities.**
(XLSX)

**S7 Table. Pairwise PERMANOVA and PERMDISP test results for arthropod ASVs (56 taxa) detected in house mouse fecal samples, based on Sørensen dissimilarities.**
(XLSX)

**S1 Appendix. Next-generation sequencing (NGS) supplemental information: PCR conditions and outcomes.**
(DOCX)

**S2 Appendix. Summary of 7-source stable isotope mixing model results, excluding mōlī (Laysan Albatross, *Phoebastria immutabilis*) as a source group.**
(DOCX)

## Acknowledgments

We acknowledge that the work described in this study was carried out on Kuaihelani of Papahānaumokuākea (Kūpuna Islands, Northwestern Hawaiian Islands), part of an indigenous space whose original people are today identified as kānaka ʻōiwi or Native Hawaiian. The authors are grateful to both Sarah Naughton and Kaylee Rosenberger for helping with house mouse hair sample preparation efforts and Amanda Adams and Elaine Beaudoin for collecting plant samples for stable isotope analysis. Megan Garfinkel, Tyler Kartzinel, and Ana Miller-ter Kuile provided helpful feedback about molecular analyses—we are very appreciative of their support. The authors thank Graham Derryberry for assistance in running models on the Rocky computer cluster at the National Institute for Mathematical and Biological Synthesis (NIMBioS) at the University of Tennessee at Knoxville. We thank Ndivhuwo Shivambu, Samuel S. Browett, and one anonymous reviewer for their comments and feedback on our manuscript. House mouse fecal and hair samples, plant samples, and arthropod specimens were collected under permit PMNM-2019-005 by the U.S. Fish and Wildlife Service (USFWS) and Island Conservation. Mōlī (Laysan Albatross) feathers were collected under permit MB56577D-0 by USFWS. We also thank Scott Shaffer, who kindly shared mōlī blood and feather samples from his research to this project (collected under permits PMNM-2009-004, PMNM-2011-015, and PMNM-2016-004). All necessary permits were obtained for the described study, which complied with all relevant regulations.

## Author Contributions

**Conceptualization:** Wieteke A. Holthuijzen, Elizabeth N. Flint, Jonathan H. Plissner, Coral A. Wolf, Holly P. Jones.

**Data curation:** Wieteke A. Holthuijzen.

**Formal analysis:** Wieteke A. Holthuijzen, Stefan J. Green, Dagmar Sweeney.

**Funding acquisition:** Wieteke A. Holthuijzen, Elizabeth N. Flint, Jonathan H. Plissner, Daniel Simberloff, Coral A. Wolf, Holly P. Jones.

**Investigation:** Wieteke A. Holthuijzen, Elizabeth N. Flint, Stefan J. Green, Jonathan H. Plissner, Dagmar Sweeney, Coral A. Wolf.

**Methodology:** Wieteke A. Holthuijzen.

**Project administration:** Wieteke A. Holthuijzen, Elizabeth N. Flint, Jonathan H. Plissner, Coral A. Wolf, Holly P. Jones.

**Resources:** Wieteke A. Holthuijzen, Elizabeth N. Flint, Stefan J. Green, Jonathan H. Plissner, Daniel Simberloff, Dagmar Sweeney, Coral A. Wolf, Holly P. Jones.

**Software:** Wieteke A. Holthuijzen.

**Supervision:** Wieteke A. Holthuijzen, Elizabeth N. Flint, Stefan J. Green, Jonathan H. Plissner, Daniel Simberloff, Dagmar Sweeney, Coral A. Wolf, Holly P. Jones.

**Validation:** Wieteke A. Holthuijzen, Elizabeth N. Flint, Stefan J. Green, Jonathan H. Plissner, Daniel Simberloff, Dagmar Sweeney, Coral A. Wolf, Holly P. Jones.

**Visualization:** Wieteke A. Holthuijzen.

**Writing – original draft:** Wieteke A. Holthuijzen.

**Writing – review & editing:** Wieteke A. Holthuijzen, Elizabeth N. Flint, Stefan J. Green, Jonathan H. Plissner, Daniel Simberloff, Dagmar Sweeney, Coral A. Wolf, Holly P. Jones.

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
