## [Decision Letter · Decision Letter 0]

28 Jun 2023

PONE-D-23-12674An invasive appetite: combining molecular and stable isotope analyses to reveal the diet of introduced house mice (Mus musculus) on a small, subtropical islandPLOS ONE

Dear Dr. Holthuijzen,

Thank you for submitting your manuscript to PLOS ONE. After careful consideration, we feel that it has merit but does not fully meet PLOS ONE’s publication criteria as it currently stands. Therefore, we invite you to submit a revised version of the manuscript that addresses the points raised during the review process.

We look forward to receiving your revised manuscript.

Kind regards,

Jianhong Zhou

Staff Editor

PLOS ONE

Reviewers' comments:

Reviewer's Responses to Questions

**Comments to the Author**

1. Is the manuscript technically sound, and do the data support the conclusions?

Reviewer #1: Yes

Reviewer #2: Yes

Reviewer #3: Yes

2. Has the statistical analysis been performed appropriately and rigorously? 

Reviewer #1: Yes

Reviewer #2: Yes

Reviewer #3: Yes

3. Have the authors made all data underlying the findings in their manuscript fully available?

Reviewer #1: Yes

Reviewer #2: Yes

Reviewer #3: No

4. Is the manuscript presented in an intelligible fashion and written in standard English?

Reviewer #1: Yes

Reviewer #2: Yes

Reviewer #3: Yes

5. Review Comments to the Author

Reviewer #1: I have reviewed a manuscript titled “An invasive appetite: combining molecular and stable isotope analyses to reveal the diet of introduced house mice (Mus musculus) on a small, subtropical island”. The authors used stable isotope and next generation sequencing to determine the diet of mice in the Sand Island. I find this study interesting and well-articulated; the introduction, methods, results, and discussion are well written. House mice are one of the most problematic invasive species, and therefore they should be removed to restore natural ecosystems in areas they have invaded. Overall, this study comprehends useful and replicable information usable at the national and international level. I have added minor edits and comments on the attached PDF. Please also add main study conclusions after recommendations.

Reviewer #2: A very interesting sound and well written paper. Amazing the number of mice you trapped with 40 traps. I very much liked the idea that the impact of eradicating an invader may be negative, and that the decision should consider such negative impacts. In the case of sea birds on Sand Island my personal conclusion from your results is that predation of birds by mice is so marginal that its populations should be monitored and surveyed regularly, but not eradicated.

Minor comments:

- « humanely euthanaised »: killing animals is never human, you follow ethical regulations that’s all, I advise that you remove this qualification from your text.

- obtained 805 pair colon hair but used only 606 selected 318 paired colon and hair samples randomly: why did you euthanize more animals than necessary? And What is the rationale behind the random selection of 318 samples

-What is the rationale behind performing analysis without and with moli?

- Variation composition of mouse diet is at the individual level: please clarify your results suggest a significant habitat and trapping effect despite high variance between individuals.

- Despite predation by mice, arthropod abundance is similar in sand island that in an island with no mice. But may be without mice predation their abundance would have been higher on sand island? Should not it be an argument in favour of not eradicating the mice?

- Preference for non-native or opportunistic behaviour? What allows you to deduce preference from your data?

- Limitation of method to detect sea bird very degraded DNA: but was not it the same problem for detecting and identifying arthropods? Your non detection might also be due to a very rare predation by mice?

- Fig 3a : not clear it gives the impression that you took hair from different part of the mouse body and that it reflects different diet

Reviewer #3: This is a really interesting study highlighting that upcoming control efforts on invasive house mice may have downstream negative effects on non-native arthropods and plant life. I was also horrified to learn about mice attacking adult albatross!

The study itself is well put together. There are some restrictions, such as sampling prey items roughly 1-2 years after the faeces were collected. I do appreciate that much of these limitations have been recognised and discussed. It makes it clear to the reader and it’s a breadth of fresh air reading a paper like this. Recognise the strengths and weaknesses. Well done.

The majority of the interpretations seem based off the stable isotopes, with some slight reinforcement from the metabarcoding. Regardless, to fully interpret the NGS data, much more info needs to be provided on the sequencing results. Poor sequencing can have a strong effect on the final results of a study. We need the details I’ve spoken about down below. Read distributions, filtering of samples, how much host DNA there was etc etc. For example, if a sample only has 100 reads after a NOVAseq run, this is unreliable. These are important details for a NGS study. I have therefore opted for major revision as these details are key. Therefore, if these are available, it should be a simple task for the authors to provide these (in supp info is perfect).

Really nice study that I’d love to see put out there, we just need more info before fully agreeing with sequencing results.

See below for specific comments:

Line 50: How will it affect sea birds? You’ve already highlighted that this may be secondary predation.

Lines 59 – 61: Remove ‘Although small and cryptic’ and change prey to predate

Lines 82 – 86: “While these studies…” This is a hefty sentence and hard to read. I’d recommend breaking this sentence down.

Line 86: Are mice understudied? They’re a common enough study species. Maybe this particular population? If so, specify this. Mice in general are well studied.

Line 89: “Pairing environmental DNA…”

Metabarcoding necessitates NGS, so pairing isn’t an ‘option’. This statement is effectively incorrect.

Additionally, please don’t state eDNA here. Animals have been killed for dietary analysis, therefore it is not eDNA. eDNA is a non-invasive approach.

Line 145: How were they euthanised?

Lines 150 – 156: This is a difficult paragraph to interpret how the samples were split/distributed. It should also specify the rationale behind decisions made. Why were females in reproductive state not included? I’m not sure what you mean by pooling males and females during the spring/summer. This could skew results? Please include a table that gives numbers of samples from each group/cohort.

Lines 168 – 172: Faeces were removed from colons stored at -20, then stored at -80 again? Why not extract DNA at time of removal?

Line 177: Why were these arthropod primers used? These are designed for eDNA sampling techniques and don’t account for host DNA being present. Authors should have explored primer options used typically in dietary studies. Especially considering a NOVAseq was used. Multiple primers could have been incorporated at this read depth.

Lines 183: The rationale behind unsuccessful bird DNA amplification is vague and unconvincing. Which primers were used? If they’re truly bird specific, then host/microbial DNA shouldn’t be an issue. Need more details for support here.

Line 189: How many cycles for the V3 kit? Any more details on the NOVAseq run?

Line 205: What criteria were used for the blast results? What percentage identity is needed for each taxonomic level?

Lines 238: This is a difficult situation. I’m more understanding of plant collection, but pitfall traps could certainly have been performed at time of mouse trapping.

Line 287: Is this not POO? Percentage of Occurrence

Line 334: These are results, not methods. Are hollow circles individual mice?

Line 370: To reasonably assess the accuracy of NGS results, we need to see read numbers. No idea on the sequencing success, raw values, the percentage of reads accounted to host, distribution of reads between individuals. This must be included. Otherwise the filtering criteria could also be skewed. Were any individuals filtered out because of low read count?

Figure 4; Why are habitat and trapping session shown? It looks like there is some clustering into ~4 groups with no explanation.

Table 3: No R values? These are typically shown with PERMANOVAs to demonstrate the amount of variance between samples explained by that variable.

Discussion is good. It’s nice to see a section acknowledging the limitations

6. PLOS authors have the option to publish the peer review history of their article (what does this mean?). If published, this will include your full peer review and any attached files.

Reviewer #1: **Yes: **Ndivhuwo SHivambu

Reviewer #2: No

Reviewer #3: **Yes: **Samuel S Browett

---

## [Author Response · Author response to Decision Letter 0]

11 Aug 2023

Reviewer 1: Ndivhuwo Shivambu

1. I have reviewed a manuscript titled “An invasive appetite: combining molecular and stable isotope analyses to reveal the diet of introduced house mice (Mus musculus) on a small, subtropical island”. The authors used stable isotope and next generation sequencing to determine the diet of mice in the Sand Island. I find this study interesting and well-articulated; the introduction, methods, results, and discussion are well written. House mice are one of the most problematic invasive species, and therefore they should be removed to restore natural ecosystems in areas they have invaded. Overall, this study comprehends useful and replicable information usable at the national and international level. I have added minor edits and comments on the attached PDF. Please also add main study conclusions after recommendations.

Thank you for your thoughtful and kind comments. We reviewed your edits and comments in the attached pdf and have updated our revised manuscript accordingly. In addition, we have decided to keep our main conclusions at the start of the discussion (as per the original manuscript) rather than adding them again after the recommendations to avoid redundancy. We are grateful for your feedback.

Reviewer 2: Anonymous

1. A very interesting sound and well written paper. Amazing the number of mice you trapped with 40 traps. I very much liked the idea that the impact of eradicating an invader may be negative, and that the decision should consider such negative impacts. In the case of sea birds on Sand Island my personal conclusion from your results is that predation of birds by mice is so marginal that its populations should be monitored and surveyed regularly, but not eradicated.

Thank you. We greatly appreciate your comments and feedback.

2. « humanely euthanaised »: killing animals is never human, you follow ethical regulations that’s all, I advise that you remove this qualification from your text.

Thank you for this making this important point. We have removed the word “humanely” from our manuscript. As a note, mice were captured and handled according to the guidelines of the American Society of Mammalogists (Sikes et al. 2011).

Reference:

Sikes RS, Gannon WL, The Animal Care and Use Committee of the American Society of Mammalogists. Guidelines of the American Society of Mammalogists for the use of wild mammals in research. J Mammal. 2011;92:235–53.

3. obtained 805 pair colon hair but used only 606 selected 318 paired colon and hair samples randomly: why did you euthanize more animals than necessary? And What is the rationale behind the random selection of 318 samples

Thank you for bringing up this point for clarification in our manuscript. Mice captured for this particular study were obtained from a larger survey (conducted by the U.S. Fish and Wildlife Service and Island Conservation) on the relative abundance and density of mice among different habitat types. We realize that this may cause confusion, so we clarified and updated the information in our manuscript (lines 156-164). For our study (and not the broader survey), we trapped 806 house mice between April 2018 to May 2019 from which we obtained both colon and hair samples. Out of these 806 mice, 606 (75.2%) mice composed the sampling pool of adult female and male mice. Specifically, we excluded juvenile mice, mice of an unknown sex, and female mice in a reproductive state (i.e., lactating, pregnant, contained embryos, etc.). We sought a sample size of ≥10 mice per habitat type, per trapping session to conduct robust statistical analyses (i.e., 10 mice x 4 habitat types x 7 trapping sessions = 280 mice). We used this number (280 mice) as the bare minimum for our sample size. To maximize our coverage of potential spatial and temporal variation in our diet, we randomly selected 318 mice—the greatest number of mice that we were able to process based on logistical and financial constraints. We have also included an additional supplementary table (S2 Table) that provides a summary of all mice captured (i.e., our sampling pool) and the subset of mice used for next-generation sequencing and stable isotope analysis. 

4. What is the rationale behind performing analysis without and with moli?

Thank you for asking this question. By using next-generation sequencing, we sought to identify and confirm which taxa are consumed by house mice on Sand Island. We then collected specimens of commonly consumed taxa (i.e., those that occur in ≥5% of sampled mice) for stable isotope analysis. However, we were unable to amplify seabird (mōlī / Laysan Albatross) DNA in mouse fecal samples (see S1 Appendix)—despite the well-documented predatory and consumptive behavior of mice on Sand Island (Duhr et al. 2019). Consequently, we decided to run two different stable isotope models. In our manuscript, we include a model with mōlī as a source group (given the predatory behavior of mice) and a model without mōlī (to accurately reflect our original approach—to only include prey items that have been identified via NGS). These models performed very similarly (lines 390-397, 454-462). When we excluded mōlī as a source group, the contribution of arthropods to mouse diet increased to 73% (from 62%), with C3 and C4 plants providing essentially the same proportion (14% and 13%, respectively—in comparison to 15% and 12% previously with mōlī as a source group; S2 Appendix). 

Reference:

Duhr M, Flint EN, Hunter SA, Taylor RV, Flanders B, Howald GR, et al. Control of house mice preying on adult albatrosses at Midway Atoll National Wildlife Refuge. In: Veitch CR, Clout MN, Martin AR, Russell JC, West CJ, editors. Island invasives: Scaling up to meet the challenge. Gland, Switzerland: International Union for Conservation of Nature; 2019. p. 21–5.

5. Variation composition of mouse diet is at the individual level: please clarify your results suggest a significant habitat and trapping effect despite high variance between individuals.

When analyzing the composition (and variability) of plant and arthropod consumption by mice, we employed two complementary multivariate analyses: PERMANOVA and PERMDISP. As we explain in our manuscript, significant results from a PERMANOVA test indicate differences in plant or arthropod consumption among groups (i.e., centroid location effect) and/or variability within groups (i.e., dispersion effect; Anderson & Walsh 2013). As a follow-up, we used the PERMDISP test to check for differences in the homogeneity of multivariate dispersion among habitat type and trapping session groups (Anderson et al. 2008); in other words, PERMDISP tests the variability of plant and arthropod composition in diet among groups. Significant results from PERMDISP help to elucidate if plant or arthropod consumption differs owing to distinct diet composition and/or variability in diet composition.

Put simply, plant consumption by mice differs significantly both in terms of composition and variability thereof among habitat types and trapping sessions, whereas arthropod consumption only differs (weakly) among habitats and sessions. However, much of the variation in mouse diet is at the individual level, which is shown in Table 3. For multivariate analyses, it is especially important to consider Pseudo F statistics and components of variation, which we report in Table 3 and discuss in our manuscript. Components of variation (described in the columns “Component”, “Var”, and “SD” of Table 3) are useful for comparing the relative importance of different factors in a model towards explaining overall variation (Anderson et al. 2008; Underwood & Petraitis 1993). More specifically, components of variation are measures of variability from a partitioning on the basis of the dissimilarity measure selected (in our case, the Sørensen dissimilarity index; Anderson et al. 2008). The estimates of the components of variation are calculated as squared units of the dissimilarity measure (“Var”); therefore, to interpret the components of variation into the original units, it is necessary to take the square root of the estimates (“SD”). For example, in our examination of the components of variation for plant diet composition in Table 3, mice from different habitat types are nearly 36% dissimilar (based on Sørensen dissimilarity units) in terms of their plant diet composition. However, at the individual level (Residual), most of the differences in diet composition are prevalent at the individual level (42% dissimilar). 

We have revised the results section in our manuscript to better clarify our results and how components of variation can be interpreted (lines 454-508). 

References:

Anderson MJ, Gorley RN, Clarke KR. PERMANOVA+ for PRIMER: Guide to software and statistical methods. Plymouth, United Kingdom: PRIMER-E Ltd; 2008.

Anderson MJ, Walsh DCI. PERMANOVA, ANOSIM, and the Mantel test in the face of heterogeneous dispersions: What null hypothesis are you testing? Ecol Monogr. 2013;83:557–74.

Underwood AJ, Petraitis PS. Structure of intertidal assemblages in different locations: how can local processes be compared? pp. 38-51 in Ricklefs RE & Schluter D (eds). Species diversity in ecological communities: historical and geographical perspectives. Chicago, Illinois, United States; University of Chicago Press; 1993.

6. Despite predation by mice, arthropod abundance is similar in sand island that in an island with no mice. But may be without mice predation their abundance would have been higher on sand island? Should not it be an argument in favour of not eradicating the mice?

Thank you for your comment. In another study (Holthuijzen et al. 2021), we conducted a baseline survey of arthropod community structure and diversity, comparing Sand Island (mice present) to Eastern Island (mice absent). Although the abundances of different arthropod taxa differ between the islands (with and without mice), it is not possible to directly correlate arthropod abundance to the presence (or absence) of mice. Sand and Eastern Islands have both different habitat types and distinct histories of land use (as well as introductions of non-native taxa). If mice are removed from Sand Island, arthropod communities (in terms of abundance, composition, and structure) may change; post-eradication monitoring is crucial to document these biotic responses. It is also important to note—as we do in our manuscript—that the vast majority of arthropod species across Kuaihelani are non-native (upwards of 80%; Nishida & Beardsley, 2002; Holthuijzen 2022). Given that house mice consume many non-native arthropods, these particular taxa may be released from predation pressure following mouse removal. However, our manuscript only provides a description of mouse diet as it pertains to the ongoing restoration and recovery of Kuaihelani; we do not make a judgement regarding the eradication of invasive house mice from Sand Island. Clearly, seabirds will benefit from the eradication of a terrestrial, non-native predator. Yet, as we state in our manuscript, it is unknown how the broader ecosystem will respond. Here, we emphasize that implementing pre- and post-eradication monitoring and management efforts could be key to mitigating “surprise effects” or adverse outcomes.

References:

Holthuijzen WA. Midway Atoll terrestrial arthropod list. Digital database. Midway Atoll National Wildlife Refuge, Honolulu, Hawaiʻi: U.S. Fish and Wildlife Service; 2022.

Holthuijzen WA, Durham SL, Flint EN, Plissner JH, Rosenberger KJ, Wolf CA, et al. Fly on the wall: Comparing arthropod communities between islands with and without house mice (Mus musculus). Pac Sci. 2021;75:371–94.

Nishida GM, Beardsley JW. A review of the insects and related arthropods of Midway Atoll. Bish Mus Occas Pap. 2002;68:25–69. 

7. Preference for non-native or opportunistic behaviour? What allows you to deduce preference from your data?

Thank you for your questions. In our manuscript, we are very cautious when describing the results of our mouse diet study. What mice consume versus what they prefer are two different dietary questions. For our manuscript, we focus on diet composition. The only instance where we speculate about mouse preference for specific food sources is in relation to arthropods in our discussion (lines 588-590). Here, we note that arthropod consumption by mice on Sand Island varies minimally throughout the year; similarly, arthropod abundance also remains relatively stable across seasons (sensu Holthuijzen et al. 2021). In turn, we hypothesize that because arthropods are abundant and accessible year-round, they may be eaten by mice owing to preference and not because of seasonal availability (as documented by Copson [1986] on Macquarie Island). However, we do not test this claim in our paper; we only provide a potential explanation for this particular trend in mouse diet (but could be tested with post-eradication arthropod monitoring). 

Given our study objectives and methods, it is not possible to clearly disentangle opportunistic behavior from a dietary preference for non-native taxa by mice; we are only able to report the composition of mouse diet and we interpret that in the context of the Sand Island ecosystem. Because most arthropod species across Kuaihelani are non-native (upwards of 80%; Nishida & Beardsley 2002; Holthuijzen 2021) it is logical to assume that mice may also consume many non-native species, whether that be due to opportunistic behavior or dietary preference. Put simply, mice seem to eat what is available and accessible; in the case of Sand Island, with such a high proportion of non-native taxa, it is not necessarily surprising to also find many non-native and invasive taxa in mouse diet. As we explain in our manuscript, mice are omnivorous, opportunistic species—which is a major reason why house mice are such widespread invasive species. Indeed, plasticity in diet is one of the key adaptations that allows house mice to survive and thrive on islands (Angel et al. 2009; Phifer-Rixey & Nachman 2015). 

References:

Angel A, Wanless RM, Cooper J. Review of impacts of the introduced house mouse on islands in the Southern Ocean: Are mice equivalent to rats? Biol Invasions. 2009;11:1743–54.

Copson GR. The diet of the introduced rodents Mus musculus L. and Rattus rattus L. on subantarctic Macquarie Island. Aust Wildl Res. 1986;13:441–5.

Holthuijzen WA. Midway Atoll terrestrial arthropod list. Digital database. Midway Atoll National Wildlife Refuge, Honolulu, Hawaiʻi: U.S. Fish and Wildlife Service; 2022.

Holthuijzen WA, Durham SL, Flint EN, Plissner JH, Rosenberger KJ, Wolf CA, et al. Fly on the wall: Comparing arthropod communities between islands with and without house mice (Mus musculus). Pac Sci. 2021;75:371–94.

Nishida GM, Beardsley JW. A review of the insects and related arthropods of Midway Atoll. Bish Mus Occas Pap. 2002;68:25–69. 

Phifer-Rixey M, Nachman MW. (2015). Insights into mammalian biology from the wild house mouse Mus musculus. ELife. 2015;4:e05959.

8. Limitation of method to detect sea bird very degraded DNA: but was not it the same problem for detecting and identifying arthropods? Your non detection might also be due to a very rare predation by mice?

Thank you for asking these questions about DNA amplification of different taxa. As we mentioned in the manuscript (and further explained in S1 Appendix), there are several reasons why amplifying seabird DNA may be challenging. Although we were unable to amplify bird DNA in the mouse fecal samples, there are likely several competing factors that affected our efforts. Firstly, fecal samples contain an overabundance of host (mouse) DNA in relation to target DNA, as well as gut microbiome DNA (Alberdi et al. 2019). In addition, there may be challenges in using COI primers targeting very degraded bird DNA versus high quality mouse DNA, especially given that both of these organisms are eukaryotic. Soft tissue (such as epithelial or muscle tissue, or blood) are more easily digestible and assimilated than harder substances (i.e., plant fibers and chitin) and therefore are likely less abundant and/or of a lower quality in fecal samples (see Larsen et al. 2022). Despite using a variety of methods to mitigate these issues of high host DNA/little target DNA (including different COI primer sets, PNA blockers, restriction enzymes, and targeted qPCR assays), we still were unable to amplify bird DNA. The consumption of seabird tissue may be rare by mice; but, given the documented predatory behavior of mice on Sand Island and abundance of seabird carcasses, it is likely that mice eat some seabird tissue. Indeed, other studies have failed to amplify bird DNA in fecal samples (e.g., Bonin et al. 2020), so we believe that this is an important knowledge gap that should be addressed in future studies. 

References:

Alberdi A, Aizpurua O, Bohmann K, Gopalakrishnan S, Lynggaard C, Nielsen M, et al. Promises and pitfalls of using high-throughput sequencing for diet analysis. Mol Ecol Resour. 2019;19:327–48. 

Bonin M, Côté SD, Dussault C, Taillon J, Lecomte N. Combining stable isotopes, morphological, and molecular analyses to reconstruct the diet of free-ranging consumers. Ecol Evol. 2020;10:6664–76.

Larsen T, Fernandes R, Wang YV, Roberts P. Reconstructing hominin diets with stable isotope analysis of amino acids: New perspectives and future directions. BioScience. 2022;72:618–37.

9. Fig 3a : not clear it gives the impression that you took hair from different part of the mouse body and that it reflects different diet

Thank you for raising this point. As we explain in the revised manuscript (lines 150-151), we only obtained hair samples from the lower abdomen area of house mice for stable isotope analysis. Figure 3A is a visual representation of the dietary proportion of four food source groups for house mouse diet overall.

Reviewer 3: Samuel S. Browett

1. This is a really interesting study highlighting that upcoming control efforts on invasive house mice may have downstream negative effects on non-native arthropods and plant life. I was also horrified to learn about mice attacking adult albatross! The study itself is well put together. There are some restrictions, such as sampling prey items roughly 1-2 years after the faeces were collected. I do appreciate that much of these limitations have been recognised and discussed. It makes it clear to the reader and it’s a breadth of fresh air reading a paper like this. Recognise the strengths and weaknesses. Well done. The majority of the interpretations seem based off the stable isotopes, with some slight reinforcement from the metabarcoding. Regardless, to fully interpret the NGS data, much more info needs to be provided on the sequencing results. Poor sequencing can have a strong effect on the final results of a study. We need the details I’ve spoken about down below. Read distributions, filtering of samples, how much host DNA there was etc etc. For example, if a sample only has 100 reads after a NOVAseq run, this is unreliable. These are important details for a NGS study. I have therefore opted for major revision as these details are key. Therefore, if these are available, it should be a simple task for the authors to provide these (in supp info is perfect). Really nice study that I’d love to see put out there, we just need more info before fully agreeing with sequencing results.

Thank you. We appreciate your detailed comments and feedback. We have thoroughly revised our manuscript; detailed information about the NGS data was described in S1 Appendix. In addition, our NGS and SIA data were available via Dryad.

2. Line 50: How will it affect sea birds? You’ve already highlighted that this may be secondary predation.

Thank you. We have updated this sentence for greater clarity in our revised manuscript (line 51).

3. Lines 59 – 61: Remove ‘Although small and cryptic’ and change prey to predate

Thank you for this comment. We have changed this text in our revised manuscript (line 59). We did not change “prey upon” to “predate” because “predate” means to exist or occur at a date earlier than something else.

4. Lines 82 – 86: “While these studies…” This is a hefty sentence and hard to read. I’d recommend breaking this sentence down.

We appreciate your feedback on this sentence and have revised it for easier reading (lines 82-86). 

5. Line 86: Are mice understudied? They’re a common enough study species. Maybe this particular population? If so, specify this. Mice in general are well studied.

Thank you for bringing up this point. Indeed, house mice are among the most popular biological models used to study a variety of traits and diseases. However, mice are specially bred and used for scientific research, notably via inbred strains (i.e., DBA; Phifer-Rixey & Nachman 2015) which greatly differ from their wild counterparts. In turn, there are large knowledge gaps about the biology and ecology of introduced, insular populations of house mice (see Angel et al. 2009). We have clarified this statement in our revised manuscript (line 87).

References:

Angel A, Wanless RM, Cooper J. Review of impacts of the introduced house mouse on islands in the Southern Ocean: Are mice equivalent to rats? Biol Invasions. 2009;11:1743–54.

Phifer-Rixey M, Nachman MW. Insights into mammalian biology from the wild house mouse Mus musculus. eLife. 2015;4:e05959. 

6. Line 89: “Pairing environmental DNA…” Metabarcoding necessitates NGS, so pairing isn’t an ‘option’. This statement is effectively incorrect. Additionally, please don’t state eDNA here. Animals have been killed for dietary analysis, therefore it is not eDNA. eDNA is a non-invasive approach.

Thank you for catching this error in our manuscript; we have revised that sentence so it now reads correctly (lines 89-92). 

7. Line 145: How were they euthanised?

Thank you for your question. As described in lines 150 and 170-176 of the revised manuscript, house mice were euthanized via cervical dislocation (per the guidelines of the American Society of Mammalogists; Sikes et al. 2011).

Reference:

Sikes RS, Gannon WL, The Animal Care and Use Committee of the American Society of Mammalogists. Guidelines of the American Society of Mammalogists for the use of wild mammals in research. J Mammal. 2011;92:235–53.

8. Lines 150 – 156: This is a difficult paragraph to interpret how the samples were split/distributed. It should also specify the rationale behind decisions made. Why were females in reproductive state not included? I’m not sure what you mean by pooling males and females during the spring/summer. This could skew results? Please include a table that gives numbers of samples from each group/cohort.

Thank you for bringing up this point for clarification in our manuscript; we have revised the text accordingly (lines 156-164). Mice captured for this particular study were obtained from a larger survey (conducted by the U.S. Fish and Wildlife Service [USFWS] and Island Conservation [IC]) on the relative abundance and density of mice among different habitat types. We realize that this may cause confusion, so we clarified and updated the information in our manuscript. For our study (and not the broader survey), USFWS and IC trapped 806 house mice between April 2018 to May 2019, from which we received both colon and hair samples. Out of these 806 mice, 606 (75.2%) mice composed the sampling pool of adult female and male mice. Specifically, we excluded juvenile mice, mice of an unknown sex, and female mice in a reproductive state (i.e., lactating, pregnant, contained embryos, etc.). We elected to exclude females in a reproductive state to reduce variation in diet caused by reproduction needs, rather than prey availability and accessibility. Reproduction and lactation impose a different and heavier nutritional burden on female mice and—in turn—influence female nutrient requirements (see NRC 1995). As indicated in the manuscript, we sought a sample size of ≥10 mice per habitat type, per trapping session to conduct robust statistical analyses (i.e., 10 mice x 4 habitat types x 7 trapping sessions = 280 mice). We used this number (280 mice) as the bare minimum for our sample size. To maximize our coverage of potential spatial and temporal variation in diet, we sampled 318 mice for diet analyses—the greatest number of mice that we were able to process based on logistical and financial constraints. We have also included an additional supplementary table (S2 Table) that provides a summary of all mice captured (i.e., our sampling pool) and the subset of mice used for next-generation sequencing (NGS) and stable isotope analysis (SIA). As a note, we tested for differences by sex for our NGS and SIA data sets and we did not find any statistically significant differences; therefore, pooling our samples (regardless of sex) did not skew our results.

Reference:

National Research Council (US) Subcommittee on Laboratory Animal Nutrition. Nutrient requirements of laboratory Animals. 4th ed. Washington (DC), United States: National Academies Press; 1995.

9. Lines 168 – 172: Faeces were removed from colons stored at -20, then stored at -80 again? Why not extract DNA at time of removal?

Sand Island is a remote field station and unfortunately does not have appropriate facilities available for lab procedures such as DNA extractions or PCRs. Colon samples from mice were stored in freezers on-island (at -20°C) until they could be shipped to facilities on the mainland for processing. We explain this in the manuscript (lines 151-154) and S1 Appendix.

10. Line 177: Why were these arthropod primers used? These are designed for eDNA sampling techniques and don’t account for host DNA being present. Authors should have explored primer options used typically in dietary studies. Especially considering a NOVAseq was used. Multiple primers could have been incorporated at this read depth.

Thank you for your question about primer set selection. Selecting the most appropriate primers is very challenging, especially when working with degraded DNA from fecal samples that also contain a high proportion of high-quality, host DNA. We considered many different primer sets, most of which were designed for amplifying arthropod DNA in low-quality samples, such as poorly-conserved museum specimens, environmental DNA from water samples, gut contents, and feces. Unfortunately, at the time when we began to work on this project, we were unable to find arthropod primers that would avoid the detection of rodent host DNA. We considered using the popular ZBJ-ArtF1c/ZBJ-ArtR2c primer set (from Zeale et al. 2011), as this primer set was successful in recovering prey DNA in bat fecal samples while avoiding the amplification of bat DNA. Although the ZBJ primers are commonly used for dietary studies (see Jusino et al. 2019), we were concerned that the taxonomic range of the ZBJ primers would not be appropriate given the broad diversity of arthropod species found across the Kuaihelani atoll (Nishida & Beardsley 2002). Indeed, several studies have noted the narrow taxonomic range and amplification bias of the ZBJ primers (see Brandon-Mong et al. 2015; Clarke et al., 2014; Elbrecht et al. 2019; Mallott et al., 2015). 

We first experimented with the primer set mlCOIintF/Fol-degen-rev (Leray et al. 2013; Yu et al. 2012), which showed high amplification efficiency in degraded samples for a diverse array of arthropod orders (Krehenwinkel et al. 2016). To initially test these primers, we made arthropod mock community samples containing known arthropod specimens from Sand Island and a control site (Nachusa Grasslands, Illinois, USA) and conducted PCRs with the mlCOIintF/Fol-degen-rev primers followed by sequencing. However, in these mock community samples, we barely detected DNA from specimens in the orders of Araneae, Dermaptera, and Diptera and did not detect Coleoptera at all (despite using a PNA-blocker gradient). We then decided to use an alternative primer set used in dietary analyses and biodiversity assessments: fwhF2/fwhR2n (Gibson et al. 2014; Leray et al. 2013; Vamos et al. 2017). Elbrecht et al. (2019) specifically noted that the fwhF2/fwhR2n primers could be used to amplify arthropod DNA from predators (via gut contents, for example) because these primers tended to have better taxonomy recovery than many other common primers, such as the ZBJ primers (Zeale et al. 2011). In addition, Elbrecht et al. (2019) noted that the fwhF2/fwhR2n primes detected ≥95% of taxa in arthropod mock communities; therefore, multiple primer sets to increase coverage were not necessary. 

Although we still recovered a high proportion of house mouse DNA in our fecal samples, we were successful in recovering arthropod DNA using the fwhF2/fwhR2n primer set. Our results mirror that of other rodent dietary studies; indeed, recent studies note the ongoing challenges of rodent mitochondrial DNA amplification (Klure et al. 2022). As a note, several primer sets used for amplification of DNA in low-quality samples (such as eDNA, gut contents, and feces) have many degeneracies, and PCRs are run at low-stringency conditions (e.g., low annealing temperature). Thus, regardless of the intended targets, primers will often target other organisms (even with a high rate of primer mismatch). Unfortunately, an arthropod-specific primer set that can avoid the amplification of rodent DNA remains elusive; we hope that primer sets appropriate for rodent dietary studies will continue to be explored and tested. 

References:

Brandon-Mong GJ, Hm G, Sing KW, Lee PS, Lim PE, Wilson JJ. DNA metabarcoding of insects and allies: An evaluation of primers and pipelines. Bull. Entomol. Res. 2015;105:717–727.

Clarke LJ, Soubrier J, Weyrich LS, Cooper A. Environmental metabarcodes for insects: In silico PCR reveals potential for taxonomic bias. Mol. Ecol. Resour. 2014;14:1160–1170.

Gibson J, Shokralla S, Porter TM, King I, van Konynenburg S, Janzen DH, Hallwachs W, Hajibabaei M. Simultaneous assessment of the microbiome and microbiome in a bulk sample of tropical arthropods through DNA metasystematics. PNAS. 2014;111:8007–8012.

Jusino MA, Banik MT, Palmer JM, Wray AK, Xiao L, Pelton E, Barber JR, Kawahara AY, Gratton C, Peery MZ, Lindner DL. An improved method for utilizing high-throughput amplicon sequencing to determine the diets of insectivorous animals. Mol. Ecol. Resour. 2019;11:176–190. 

Klure DM, Greenhalgh R, Dearing MD. Addressing nontarget amplification in DNA metabarcoding studies of arthropod-feeding rodets. Mamm Res. 2022;67:499–509.

Leray M, Yang JY, Meyer CP, Mills SC, Agudelo N, Ranwez V, Boehm JT, Machida RJ. A new versatile primer set targeting a short fragment of the mitochondrial COI region for metabarcoding metazoan diversity: application for characterizing coral reef fish gut contents. Front. Zoo. 2013;10:34.

Mallott E, Malhi R, Garber P. High-throughput sequencing of fecal DNA to identify insects consumed by wild Weddell's saddleback tamarins (Saguinus weddelli, Cebidae, Primates) in Bolivia. Am. J. Phys. Anthropol. 2015;156:474–481.

Nishida GM, Beardsley JW. A review of the insects and related arthropods of Midway Atoll. Bish Mus Occas Pap. 2002;68:25–69.

Vamos EE, Elbrecht V, Leese F. Short COI markers for freshwater macroinvertebrate metabarcoding. MBMG. 2017;1:e14625.

Yu DW, Ji Y, Emerson BC, Wang X, Ye C, Yang C, Ding Z. Biodiversity soup: Metabarcoding of arthropods for rapid biodiversity assessment and biomonitoring. Methods Ecol. Evol. 2012;3:613–623.

Zeale MRK, Butlin RK, Barker GLA, Lees DC, Jones G. Taxon-specific PCR for DNA barcoding arthropod prey in bat faeces. Mol. Ecol. Resour. 2011;11:236–244. 

11. Lines 183: The rationale behind unsuccessful bird DNA amplification is vague and unconvincing. Which primers were used? If they’re truly bird specific, then host/microbial DNA shouldn’t be an issue. Need more details for support here.

Please refer to S1 Appendix, where we discussed the challenges of bird DNA amplification from fecal samples in detail, including different primers that we used and developed, PNA blockers, qPCR outcomes, troubleshooting, and suggestions/best practices for future studies. In the S1 Appendix, we described the difficulties of working with fecal material (i.e., an overabundance of host [mouse] DNA in relation to target DNA, as well as gut microbiome DNA). In addition, there are ongoing challenges when using COI primers to target very degraded bird DNA versus high quality mouse DNA (especially given that both of these organisms are eukaryotic). 

12. Line 189: How many cycles for the V3 kit? Any more details on the NOVAseq run?

Thank you for these questions. For plant amplicons, DNA sequencing was performed using an Illumina MiSeq, with a flow cell employing V3 chemistry with 600 cycles (2 x 300 bp). For arthropod amplicons, DNA sequencing took place on an Illumina NovaSeq 6000 instrument with an SP flow cell type (2 x 250 bp), owing to high host DNA amplification. PCR conditions and outcomes as well as sequencing information are provided in the S1 Appendix.

13. Line 205: What criteria were used for the blast results? What percentage identity is needed for each taxonomic level?

Thank you for your questions. We generated separate amplicon sequence variant (ASV) tables for plant and arthropod amplicons using the DADA2 pipeline with a 97% identity (sequence similarity) threshold in order to increase accuracy of taxonomic assignment and exclude chimeric sequences (lines 208-223). We retained only ASVs identified at least to genus (i.e., percentage identity of ≥97% at genus and species levels; sensu Garfinkel et al. 2020; McInnes et al. 2017). Other custom parameters used to match sequences to references using nucleotide BLAST included: 

• -evalue = 0.0001

• -outfmt = '6=qseqid=sseqid=staxid=score=evalue=length=qlen=pident'

• -perc_identity = 97

• -max_target_seqs = 5

References:

Garfinkel MB, Minor ES, Whelan CJ. Birds suppress pests in corn but release them in soybean crops within a mixed prairie/agriculture system. Condor. 2020;122:1–12. 

McInnes JC, Alderman R, Lea MA, Raymond B, Deagle BE, Phillips RA, Stanworth A, Thompson DR, Catry P, Weimerskirch H, et al. High occurrence of jellyfish predation by Black-browed and Campbell albatross identified by DNA metabarcoding. Molecular Ecology. 2017;26:4831–4845.

14. Lines 238: This is a difficult situation. I’m more understanding of plant collection, but pitfall traps could certainly have been performed at time of mouse trapping.

The ability to collect prey specimens at the same time and location as samples from the target organism is indeed difficult, and we thank the reviewer for understanding these challenges. However, arthropod pitfall trapping occurred simultaneously along the same traplines used to capture house mice for this study (see lines 245-247). The arthropods captured in these pitfall traps were used as samples for SIA and our mixing model.

15. Line 287: Is this not POO? Percentage of Occurrence

Thank you for raising this point. Frequency of occurrence (FOO) is defined as the number of samples that contain a given food taxon; FOO is often expressed as a percent (%FOO), which is the metric we use in our manuscript. Percent of occurrence (POO) is %FOO rescaled so that the sum across all food items is 100%. The equations for calculating %FOO and POO are shown below (per Deagle et al. 2019). We do not report POO in our figures or tables; our tables specifically note the count (or number of samples that contain a given ASV) and the %FOO for each ASV.

Here, T is the number of food taxa, S is the number of samples, and I is an indicator function such that Ii,k =1 if the food taxon i is present in the sample k, and 0 if not.

Reference:

Deagle BE, Thomas AC, McInnes JC, Clarke LJ, Vesterinen EJ, Clare EL, et al. Counting with DNA in metabarcoding studies: How should we convert sequence reads to dietary data? Mol Ecol. 2019;28:391–406.

16. Line 334: These are results, not methods. Are hollow circles individual mice?

Per PLoS ONE guidelines, we are required to place figure captions directly after the paragraph in which they are first cited. Thank you for catching the error in our legend for Figure 2; we have updated this figure in the revised manuscript where we indicate that the hollow, grey circles represent stable isotope values of individual house mouse hair samples. 

17. Line 370: To reasonably assess the accuracy of NGS results, we need to see read numbers. No idea on the sequencing success, raw values, the percentage of reads accounted to host, distribution of reads between individuals. This must be included. Otherwise the filtering criteria could also be skewed. Were any individuals filtered out because of low read count?

We thank you for asking these important questions. Please refer to S1 Appendix, where we provided this information.

18. Figure 4; Why are habitat and trapping session shown? It looks like there is some clustering into ~4 groups with no explanation.

Per the predictions that we outlined in the beginning of our manuscript, we sought to analyze spatial (habitat type) and temporal (trapping session) variation in mouse diet composition (more specifically, plant and arthropod composition; lines 109-111). In our methods, we described the multivariate analyses that we used to test for spatial and temporal differences in plant and arthropod consumption by mice (lines 306-319). We noted that we would use principal coordinates analysis (PCoA) to visually examine the output from our two resemblance matrices of occurrence data (plant and arthropod ASVs) using the Sørensen dissimilarity index. A detailed overview of the results from multivariate analyses and Figure 4 are provided in the results section (lines 453-518). We have also revised Figure 4 so it corresponds to the order of our results (i.e., habitat type in Fig 4A and trapping session in Fig 4B). 

19. Table 3: No R values? These are typically shown with PERMANOVAs to demonstrate the amount of variance between samples explained by that variable.

Thank you for your excellent question about R2 values in relation to PERMANOVAs. For multivariate analyses, it is especially important to consider the estimated sizes of components of variation, which we report in Table 3 and discuss in our manuscript. Components of variation (described in the columns “Component”, “Var”, and “SD” of Table 3) are useful for comparing the relative importance of different effects in a model towards explaining overall variation (Anderson et al. 2008; Underwood & Petraitis 1993). More specifically, components of variation are measures of variability from a partitioning on the basis of the dissimilarity measure selected (in our case, the Sørensen dissimilarity index; Anderson et al. 2008). The estimates of the components of variation are calculated as squared units of the dissimilarity measure (“Var”); therefore, to interpret the components of variation into the original units, it is necessary to take the square root of the estimates (“SD”). 

More importantly, these estimates of components of variation should be used as the basis for comparing the relative importance of different effects in a model towards explaining overall variation (Underwood & Petraitis 1993). In contrast, the raw sums of squares (or taken as a percentage of the total sum of squares) or R2 values are not always directly comparable. This is because different effects generally have different degrees of freedom. In other words, it would be inappropriate to compare the percentage of the total sum of squares explained by habitat type (which has 3 degrees of freedom) versus trapping sessions (which has 6 degrees of freedom). R2 values can be used for simpler models (without multiple effects), but in the case of our model (with fixed effects of differing degrees of freedom), R2 values are not directly comparable among model effects. In light of these considerations, we provide the estimated sizes of the components of variation in Table 3. We have added extra details in our methods and results sections in our revised manuscript (lines 312-313, Table 3) to better clarify our statistical analyses and how components of variation can be interpreted. 

References:

Anderson MJ, Gorley RN, Clarke KR. PERMANOVA+ for PRIMER: Guide to software and statistical methods. Plymouth, United Kingdom: PRIMER-E Ltd; 2008.

Anderson MJ, Walsh DCI. PERMANOVA, ANOSIM, and the Mantel test in the face of heterogeneous dispersions: What null hypothesis are you testing? Ecol Monogr. 2013;83:557–74.

Underwood AJ, Petraitis PS. Structure of intertidal assemblages in different locations: how can local processes be compared? pp. 38-51 in Ricklefs RE & Schluter D (eds). Species diversity in ecological communities: historical and geographical perspectives. Chicago, Illinois, United States; University of Chicago Press; 1993.

20. Discussion is good. It’s nice to see a section acknowledging the limitations

Thank you for your comment about our discussion and limitations section. We wanted to provide transparency about our methods and results, as well as caveats when working with (and interpreting) sequencing and stable isotope data.

---

## [Decision Letter · Decision Letter 1]

5 Oct 2023

An invasive appetite: combining molecular and stable isotope analyses to reveal the diet of introduced house mice (Mus musculus) on a small, subtropical island

PONE-D-23-12674R1

Dear Dr. Holthuijzen,

We’re pleased to inform you that your manuscript has been judged scientifically suitable for publication and will be formally accepted for publication once it meets all outstanding technical requirements.

Kind regards,

Neelesh Dahanukar, Ph.D.

Academic Editor

PLOS ONE

Additional Editor Comments:

Authors have revised the manuscript as per the comments made on the earlier draft. The manuscript, in the current form, is both scientifically and statistically sound and can be accepted for publication.

Reviewers' comments:

Reviewer's Responses to Questions

**Comments to the Author**

1. If the authors have adequately addressed your comments raised in a previous round of review and you feel that this manuscript is now acceptable for publication, you may indicate that here to bypass the “Comments to the Author” section, enter your conflict of interest statement in the “Confidential to Editor” section, and submit your "Accept" recommendation.

Reviewer #1: All comments have been addressed

2. Is the manuscript technically sound, and do the data support the conclusions?

Reviewer #1: Yes

3. Has the statistical analysis been performed appropriately and rigorously? 

Reviewer #1: Yes

4. Have the authors made all data underlying the findings in their manuscript fully available?

Reviewer #1: Yes

5. Is the manuscript presented in an intelligible fashion and written in standard English?

Reviewer #1: Yes

6. Review Comments to the Author

Reviewer #1: The manuscript has improved so much,. Well done, I do not have any further comments on the manuscript

7. PLOS authors have the option to publish the peer review history of their article (what does this mean?). If published, this will include your full peer review and any attached files.

Reviewer #1: **Yes: **Ndivhuwo Shivambu

---

## [Editor Report · Acceptance letter]

10 Oct 2023

PONE-D-23-12674R1 

An invasive appetite: combining molecular and stable isotope analyses to reveal the diet of introduced house mice (*Mus musculus*) on a small, subtropical island 

Dear Dr. Holthuijzen:

I'm pleased to inform you that your manuscript has been deemed suitable for publication in PLOS ONE. Congratulations! Your manuscript is now with our production department. 

Kind regards, 

on behalf of

Dr. Neelesh Dahanukar 

Academic Editor

PLOS ONE